# REPORT

# LUZP1 and the tumor suppressor EPLIN modulate actin stability to restrict primary cilia formation

João Gonçalves[1], Amit Sharma[1], Étienne Coyaud[2], Estelle M.N. Laurent[2], Brian Raught[2,3], and Laurence Pelletier[1,4]

Cilia and flagella are microtubule-based cellular projections with important sensory and motility functions. Their absence or malfunction is associated with a growing number of human diseases collectively referred to as ciliopathies. However, the fundamental mechanisms underpinning cilia biogenesis and functions remain only partly understood. Here, we show that depleting LUZP1 or its interacting protein, EPLIN, increases the levels of MyosinVa at the centrosome and primary cilia formation. We further show that LUZP1 localizes to both actin filaments and the centrosome/basal body. Like EPLIN, LUZP1 is an actin-stabilizing protein that regulates actin dynamics, at least in part, by mobilizing ARP2 to the centrosomes. Both LUZP1 and EPLIN interact with known ciliogenesis and cilia-length regulators and as such represent novel players in actin-dependent centrosome to basal body conversion. Ciliogenesis deregulation caused by LUZP1 or EPLIN loss may thus contribute to the pathology of their associated disease states.

## Introduction

Cilia are microtubule (MT)–based organelles that protrude from the cell surface. In vertebrates, multiple immotile (i.e., primary cilia) and motile cilia fulfil critical sensory and motility functions required for embryonic development and adult tissue homeostasis (Goetz and Anderson, 2010; Mirvis et al., 2018; Mitchell, 2007). Defects in cilia biogenesis and functions cause human diseases typified by symptoms such as blindness, infertility, and cystic kidneys (Mitchison and Valente, 2017). Ciliogenesis is not fully understood but involves multiple cellular machineries, including the cytoskeleton, membrane traffic, and centriolar satellites (Hsiao et al., 2012; Mirvis et al., 2018; Odabasi et al., 2019). The MT and actin cytoskeletons act jointly in processes such as cell adhesion, migration, and mitotic spindle orientation (Dogterom and Koenderink, 2019) and also in ciliogenesis (Mirvis et al., 2018; Pitaval et al., 2017). Cilia biogenesis initiates at the centrosome, a MT and actin organizing center (Farina et al., 2016), and relies on its older (mother) centriole to form the basal body from which the ciliary axoneme is nucleated (Lu et al., 2015). At the onset of ciliation, a ciliary vesicle is formed at the distal end of the mother centriole/basal body, which then moves to the cell surface, where it attaches to the cell membrane through transition fibers (Gonçalves and Pelletier, 2017). This migration process relies both on increased MT polymerization at the centrosome and increased actin contractility (Pitaval et al., 2017). Interestingly, loss of function of actin

regulators or pharmacological disruption of the actin cytoskeleton (e.g. cytochalasin D treatments) increases ciliation and affects ciliary length and signaling (Kim et al., 2010; Nagai and Mizuno, 2017). Treating cells with cytochalasin D leads to the accumulation of MyosinVa at the centrosome, which promotes ciliary vesicle formation (Lu et al., 2015; Wu et al., 2018). How global and/or centrosomal actin dynamics affects cilia biogenesis is not fully understood.

Here, we show that LUZP1 localizes to the centrosome, the basal body, actin filaments, and the midbody and that loss of LUZP1 function increases ciliation in human RPE-1 (retinal pigmented epithelium) cells. Using proximity-dependent biotin identification (BioID; Roux et al., 2012), coimmunoprecipitation (coIP), and functional assays, we demonstrate that the actin stabilizing protein EPLIN interacts with LUZP1 and also restricts ciliation. We further show that LUZP1 and EPLIN modulate actin and actin-associated protein (MyosinVa and ARP2) levels at centrosomes.

## Results and discussion

We previously used BioID to characterize the centrosome–cilium interface in human cells (Gupta et al., 2015). This study identified LUZP1 as a prey for proteins that localize to centriolar satellites, the centrosome, and primary cilia, indicative of potential

[1]Lunenfeld-Tanenbaum Research Institute, Mount Sinai Hospital, Toronto, Ontario, Canada; [2]Princess Margaret Cancer Centre, University Health Network, Toronto, Ontario, Canada; [3]Department of Medical Biophysics, University of Toronto, Toronto, Ontario, Canada; [4]Department of Molecular Genetics, University of Toronto, Toronto, Ontario, Canada.

Correspondence to João Gonçalves: joao.alg@gmail.com; Laurence Pelletier: pelletier@lunenfeld.ca; J. Gonçalves's present address is Deep Genomics, MaRS Centre, Toronto, Ontario, Canada.



centrosomal/ciliary localization and function (Gupta et al., 2015).

LUZP1 contains an N-terminal LCD1 (Rouse and Jackson, 2000) and three leucine zipper domains (Fig. 1 A; Sun et al., 1996) and is predominantly expressed in the mouse brain and neural lineages (Lee et al., 2001; Sun et al., 1996). Using antibodies to LUZP1 and centrosome/cilia markers, we found that endogenous LUZP1 localizes to the centrosome, the basal body, actin fibers, and the midbody (Fig. 1, B–D). These results were confirmed by analyzing GFP-LUZP1 in fixed cells (Fig. S1, A–C) and by time-lapse imaging (Video 1). Treatment with nocodazole revealed that the protein could still localize to the centrosome in absence of MTs (Fig. S1 D). To refine the localization of LUZP1, we stained RPE-1 cells (WT and expressing GFP-LUZP1) with antibodies against different centriole and ciliary domains, including subdistal appendages (CEP170), distal appendages/transition fibers (CEP164), and the transition zone (CEP290; Figs. 1 C and S1 C). Our analysis revealed that LUZP1 localizes at the proximal end of basal bodies. To determine which protein domains within LUZP1 are important for its localization, we generated two truncation mutants, one spanning aa 1–296 and the other the remainder of the protein (aa 297–1,076; Fig. 1 A). Our results indicated that the leucine zipper and LCD1 domains are not required for LUZP1's localization (Fig. 1 E).

To assess the molecular pathways in which LUZP1 operates, we defined its proximity interactome using BioID in cycling cells and cells serum starved to promote primary cilia formation. FLAG-BirA*-LUZP1 localized to the centrosome and the actin cytoskeleton and effectively biotinylated proteins (Fig. S1 E). Statistical analyses were performed to define high-confidence proximal interactors (Fig. 2 A and Table S1). Consistent with its subcellular localization and previous BioID studies, the majority of LUZP1 preys localized to centriolar satellites, the centrosome/cilium, and the MT and actin cytoskeletons (Fig. 2 A and Table S1), suggesting that LUZP1 has cytoskeleton-related roles. We had previously identified LUZP1 as a proximity interactor of PCM1 (Gupta et al., 2015), a centriolar satellite component likely biotinylated by FLAG-BirA*-LUZP1 at the centrosome, since LUZP1 was not detected in satellite granules. PCM1 mediates the localization of actin regulators (e.g. ARP2/3 complex and Wiskott Aldrich Syndrome protein and scar homologue complex) to the centrosome and is involved in its function as an actin-organizing center (Farina et al., 2016). Myosin subunits were also identified, some of which (MYL12A/B, MYL3, or MYL6/B) were found only in serum-starved cells. Consistent with this, DAPK3, which regulates myosin by phosphorylating MYL9 and MYL12B (Komatsu and Ikebe, 2004; Murata-Hori et al., 1999), was also detected. Myosins are actin-binding molecular motors that play several roles in processes like cell migration and division (Hartman and Spudich, 2012). Of note, myosins can accumulate at the centrosome and primary cilium upon actin destabilization (Wu et al., 2018). Therefore, our results may reflect actin cytoskeleton changes happening during ciliogenesis. A number of additional actin-binding proteins were identified as putative LUZP1 interactors, including SEPT2, which localizes along the primary cilium axoneme and is required for ciliogenesis (Ghossoub et al., 2013; Hu et al., 2010); TJP1, which is

part of the tight junction structure and interacts with actin to establish cellular connections (Fanning et al., 1998); and SORBS1/CAP, which plays a role in the formation of actin stress fibers and focal adhesions and plays a role in cell adhesion, spreading, and motility (Ribon et al., 1998; Zhang et al., 2006). We also detected Filamins (FLNA/B), which play important actin-related functions by linking the actin cytoskeleton network to membrane components (Nakamura et al., 2011), as LUZP1 preys consistent with a previous report (Wang and Nakamura, 2019). Of most interest to us was the observation that EPLIN/LIMA1 was also detected as a LUZP1 prey. EPLIN binds to, cross-links, and stabilizes actin filaments (Maul et al., 2003). In addition, it plays important roles at cellular junctions and focal adhesions (Abe and Takeichi, 2008; Karaköse et al., 2015). LUZP1 was identified as an EPLIN interactor by affinity purification followed by mass spectrometry (MS) to identify the interactome of over 1,000 human proteins in HeLa cells (Hein et al., 2015). Also, both LUZP1 and EPLIN are part of the BioID interactome of the human phosphatase CDC14A, which modulates actin through EPLIN dephosphorylation (Chen et al., 2017) and regulates actin nucleation at the centrosome, impacting primary cilia length (Uddin et al., 2019). These orthogonal MS approaches supported the notion that EPLIN and LUZP1 associate functionally, a possibility we sought to pursue.

Two alternative promoters drive the expression of the *LIMA1* gene, resulting in the expression of EPLIN isoforms α and β (Fig. 2 B; Chen et al., 2000). Both isoforms contain a central LIM domain possibly involved in protein–protein interactions (Maul and Chang, 1999). Also, EPLIN proteins have at least two actin-binding domains (Maul et al., 2003). EPLINα and β are differently expressed in a number of cell lines and tissues (Maul and Chang, 1999; Maul et al., 2001; Wang et al., 2007), with RPE-1 cells expressing mostly EPLINα (Fig. 2 C). To validate the interaction between LUZP1 and EPLIN, we performed semi-endogenous coIP experiments using RPE-1 cells stably expressing GFP-tagged LUZP1 or EPLIN and observed that GFP-tagged LUZP1 pulled down both endogenous EPLIN isoforms (Fig. 2 D). Similarly, GFP-tagged EPLINα and β pulled down endogenous LUZP1 (Fig. 2 E). We further showed that FLAG-tagged EPLINβ pulled down endogenous LUZP1 in HEK293 cells, indicating the observed interactions were not dependent on the tag used for the coIPs (Fig. S2 A). To determine if LUZP1 leucine zipper and LCD1 domains are important for the interaction with EPLIN, we expressed GFP-EPLINβ in HEK293 stable/inducible lines expressing different constructs: FLAG-LUZP1 (full length), FLAG-LUZP1 (aa 1–296), or FLAG-LUZP1 (aa 297–1,076). By coIP, we observed that GFP-EPLINβ pulled down only full-length FLAG-LUZP1 and FLAG-LUZP1 (aa 297–1,076) showing that the C-terminal region of LUZP1 mediates the interaction with EPLIN (Fig. S2 B). Finally, we also showed that FLAG-LUZP1 and FLAG-EPLINβ pull down actin (Fig. S2, A and C).

To study the localization of EPLIN in RPE-1 cells and compare it to that of LUZP1, we costained cells with antibodies specific to both proteins and observed them localizing at actin filaments. However, whereas EPLIN accumulated at the leading edge of the cell, localizing at actin ruffles, LUZP1 was predominantly at the opposite end of the cell (Fig. 2 F). This shows that even though LUZP1 and EPLIN colocalize at a subset of actin filaments, they

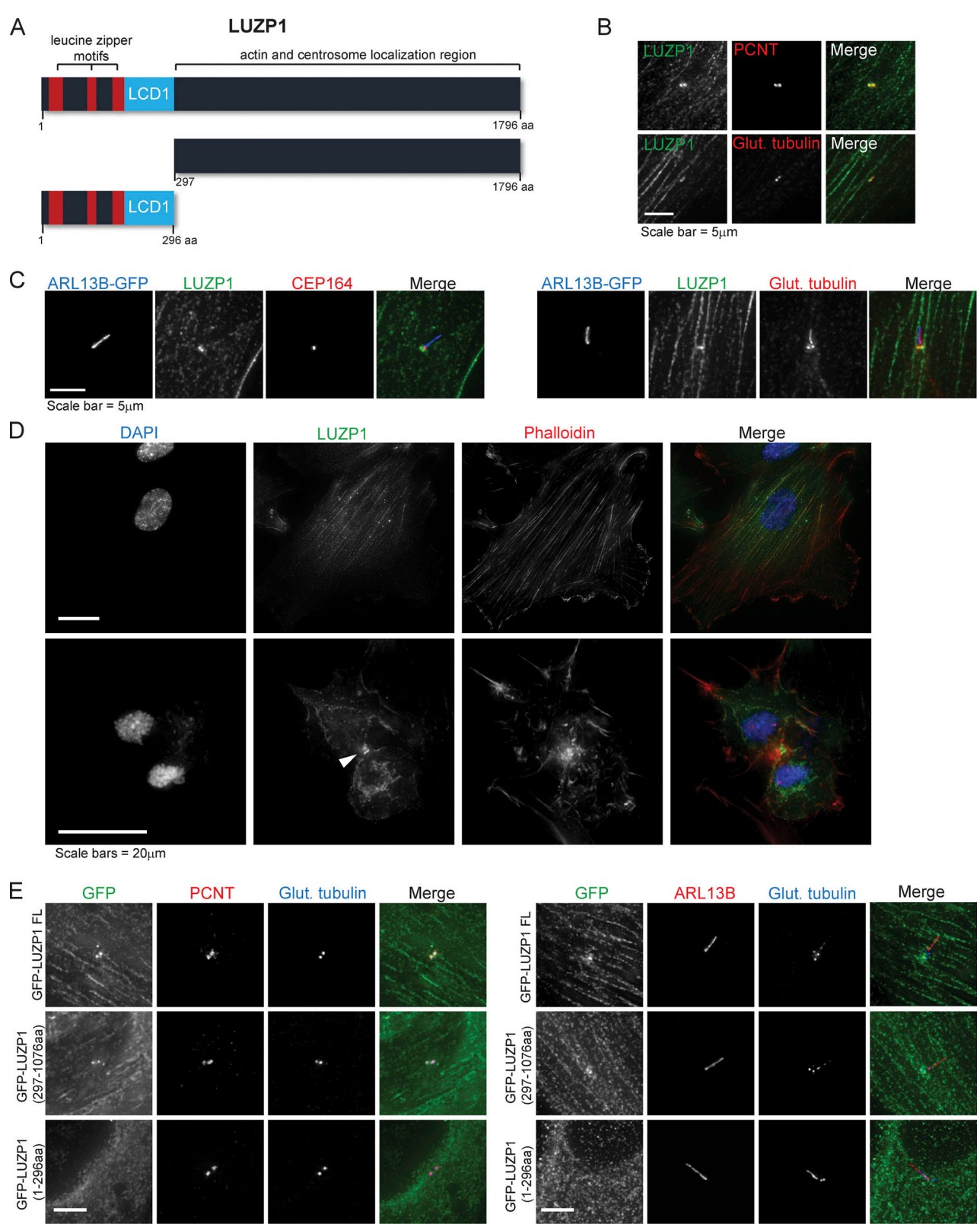

Figure 1. **Subcellular localization of LUZP1, a novel centrosome/basal body and actin-associated protein. (A)** Schematic representation of human LUZP1, a 1,796-aa protein containing three leucine-zipper motifs and a LCD1 domain at its N-terminal region. Represented are also two truncation mutants. **(B)** LUZP1 localizes to the centrosome. RPE-1 cells were stained for LUZP1 and centrosome markers PCNT (pericentriolar material marker) and glutamylated tubulin (centriole marker). **(C)** LUZP1 localizes to the basal body. RPE-1 cells stably expressing ARL13B-GFP were stained for GFP, LUZP1, CEP164 (distal appendage/transition fiber marker), and glutamylated tubulin (centrosome and ciliary marker). **(D)** LUZP1 localizes to actin filaments and the midbody (indicated by the arrowhead). RPE-1 cells were stained using an antibody against LUZP1. The actin cytoskeleton was stained with fluorophore-conjugated

phalloidin, and DNA was stained with DAPI. **(E)** The N-terminal region of LUZP1 is not required for the protein's localization. RPE-1 stable cells stably expressing GFP-tagged LUZP1, full-length and the two truncation mutants represented in A were stained for GFP, glutamylated tubulin (centriole and cilium marker), and PCNT (a centrosome marker; left panels) or ARL13B (right panels).

accumulate at distinct actin structures, suggesting they may play different roles in actin-related processes such as cell migration. To study the localization of each EPLIN isoform in RPE-1 cells, we used lines stably expressing each GFP-tagged EPLIN isoform alone or in combination with mCherry-LUZP1 (Figs. 2 F and S1 F). Both EPLIN isoforms localized to actin filaments; however, isoform α tended to accumulate on filopodia and membrane ruffles at the cell cortex, often at what was clearly the leading edge of a moving cell (Fig. 2 F). These results are consistent with the localization of endogenous EPLIN and the fact that EPLINα is the major isoform in these cells. In contrast to LUZP1, none of the EPLIN isoforms were observed at the centrosome, basal body, or midbody (Fig. 2 G; and Fig. S1, F and G). It will prove interesting to investigate the localization of these proteins beyond the diffraction limit in the future.

Next, we investigated the role of LUZP1 and EPLIN by determining their loss-of-function phenotypes in RPE-1 cells. Previously, LUZP1 scored as a negative regulator of ciliogenesis in an RNAi screen of the centrosome–cilium interface proximity interactome (Gupta et al., 2015). Taking this into account, as well as LUZP1 localization to the basal body, we decided to investigate a potential role for this protein in ciliogenesis by conducting RNAi experiments. After transfection with a negative control siRNA or siRNAs targeting LUZP1, the cells were serum starved for 72 h to induce primary cilia formation. In agreement with the RNAi screen, we observed a higher number of ciliated cells in the LUZP1 RNAi condition (Fig. S2 D). To explore these results further and confirm the specificity of the phenotype, we tested if LUZP1 depletion could induce ciliogenesis in conditions that do not favor it (i.e., in the presence of serum). We also tested if the phenotype is specific by using RPE-1 cells expressing GFP-LUZP1 constructs, one RNAi sensitive and the other resistant. As controls, we used cells expressing GFP only (Fig. 3, A–C). The different cell lines were transfected with control or LUZP1 siRNAs for 72 h but were not serum starved. In these conditions, we observed a significant increase in the number of ciliated cells in the LUZP1-depleted populations, except in the cell line expressing the RNAi-resistant construct (Fig. 3, A and B). Western blot analysis confirmed the depletion of endogenous and GFP-tagged LUZP1 in the sensitive cell line but only of the endogenous protein in the resistant one (Fig. 3 C). These results show that the phenotype is specific and confirm LUZP1 as a negative regulator of primary cilia formation.

Next, we tested if a similar phenotype would be observed for EPLIN. For this assay, RPE-1 cells expressing GFP only or GFP-EPLINα were transfected with control siRNAs or siRNAs targeting the 3′ UTR of *EPLIN* transcripts (Fig. 3, D–F). The siRNAs efficiently depleted endogenous EPLIN, but not the GFP fusion (Fig. 3 F). 72 h after transfection, in the presence of serum, we observed a significant increase in the number of ciliated cells in the control cell line, but not in the one expressing GFP-EPLINα (Fig. 3, D–F). Thus, like LUZP1, EPLIN is a negative regulator of

ciliogenesis. Supporting this conclusion, upon serum starvation, LUZP1- and EPLIN-overexpressing RPE-1 lines showed a small decrease in ciliation compared with cells expressing only GFP (Fig. S2 E). To rule out cell cycle arrest as a cause for the increased ciliation in cells depleted of LUZP1 and EPLIN, we followed their growth throughout the experiment by time-lapse imaging and showed that cell proliferation was not perturbed by the RNAi of any of the genes (Fig. S2 F). Given EPLIN's actin-stabilizing function (Maul et al., 2003), LUZP1 localization to the centrosome/basal body and actin filaments, and actin-associated BioID preys, we hypothesized that affected actin-related processes might be causing the increased ciliation. Indeed, disrupting the actin cytoskeleton pharmacologically or through the depletion of actin regulators (e.g., ARP3) causes increased ciliation and primary cilia lengthening (Kim et al., 2010). Hence, we tested if ciliary length was affected in cells depleted of LUZP1 and EPLIN. By measuring cilia length in the cell populations described above, we showed that LUZP1- or EPLIN-silenced cells had significantly longer primary cilia than control cells. These phenotypes were rescued by the expression of the respective protein as a GFP fusion, confirming their specificity (Fig. 3, G and H). Given the similarity in loss-of-function phenotypes between LUZP1 and EPLIN, we posited that LUZP1 could also stabilize the actin cytoskeleton. To test this, we used an approach that supported EPLIN's actin-stabilization properties (Maul et al., 2003). Specifically, we transiently expressed FLAG-tagged LUZP1 or EPLIN in MCF-7 and RPE-1 cells and stained them for actin (Fig. 4, A and B). Compared with control cells transfected with the empty plasmid and nonexpressing neighboring cells, cells overexpressing FLAG-LUZP1 or FLAG-EPLIN showed brighter actin filaments, suggesting their stabilization (Fig. 4, A and B). Also, in agreement with a recent report showing that LUZP1 fragments (aa 1–500; aa 400–500) cross-link actin filaments in vitro (Wang and Nakamura, 2019), the overexpression of FLAG-tagged LUZP1 aa 1–496 in RPE-1 cells dramatically affected actin organization, with the filaments appearing curved likely due to bundling (Fig. 4 C). As previously mentioned, low doses of cytochalasin D, an inhibitor of actin polymerization, induce ciliation by causing the accumulation of MyosinVa at the centrosome, which promotes the formation of the ciliary vesicle. To further investigate LUZP1 potential actin stabilization role, we posited that overexpressing it could counteract the effect of cytochalasin D on ciliation. To test this hypothesis, we used two control cell lines (WT and GFP-only RPE-1 cells) and cell lines expressing GFP-LUZP1, GFP-EPLINα, GFP-EPLINβ, or the EPLIN fusions in combination with mCherry-LUZP1 (Fig. 4 D). The cells were treated with a low dose of cytochalasin D (50 nM) for 16 h in the presence of serum, after which ciliation was assessed. As expected, this caused a significant increase in the number of ciliated cells in control cells. However, in the overexpressing lines the increase in ciliation was significantly less (approximately half) than in

A

| Symbol | | Control peptide counts | LUZP1 Peptide counts | Sum | SAINT | BFDR | LUZP1 ciliated Peptide counts | Sum | SAINT | BFDR | cil/non-cil log2 fc |
|---|---|---|---|---|---|---|---|---|---|---|---|
| LUZP1 | *Leucine zipper protein 1* | 0\|0\|0\|0\|0\|0\|0\|0 | 5833\|5426\|5290 | 16549 | na | na | 4556\|4240\|4105 | 12901 | na | na | -0.36 |
| BirA | *Bifunctional ligase/repressor BirA R118G epitope tag* | 2423\|2543\|1879\|1698\|1572\|1711\|1566\|1902 | 4074\|4005\|3901 | 11980 | na | na | 3729\|3704\|3497 | 10930 | na | na | -0.13 |
| FLNA/FLNB* | *filamin A and filamin B* | 174\|176\|482\|469\|863\|876\|825\|839 | 3815\|3385\|3373 | 10573 | 1.00 | 0.00 | 3944\|3780\|3558 | 11282 | 1.00 | 0.00 | 0.09 |
| DAPK3 | *death associated protein kinase 3* | 0\|0\|0\|0\|0\|0\|0\|0 | 282\|227\|214 | 723 | 1.00 | 0.00 | 523\|500\|446 | 1469 | 1.00 | 0.00 | 1.02 |
| TUBA1C | *tubulin alpha 1c* | 124\|140\|0\|0\|159\|146\|142\|138 | 385\|337\|319 | 1041 | 0.00 | 0.48 | 409\|401\|398 | 1208 | 1.00 | 0.00 | 0.21 |
| HSPA1B | *Hsp70 member 1B* | 112\|130\|145\|138\|154\|153\|153\|143 | 475\|442\|425 | 1342 | 1.00 | 0.00 | 400\|399\|353 | 1152 | 0.67 | 0.06 | -0.22 |
| MRE11A | *MRE11 homolog, double strand break repair nuclease* | 30\|27\|45\|46\|63\|70\|65\|65 | 59\|57\|56 | 172 | 0.00 | 0.77 | 201\|185\|172 | 558 | 0.73 | 0.00 | 1.69 |
| MYH9 | *myosin heavy chain 9* | 0\|0\|0\|0\|0\|0\|0\|0 | 25\|19\|13 | 57 | 1.00 | 0.00 | 150\|148\|144 | 442 | 1.00 | 0.00 | 2.93 |
| CCT3 | *chaperonin containing TCP1 subunit 3* | 15\|10\|25\|32\|31\|26\|26\|27 | 40\|24\|18 | 82 | 0.00 | 0.74 | 109\|103\|90 | 302 | 1.00 | 0.00 | 1.87 |
| MYL12A/B** | *myosin light chain 12A/B* | 0\|0\|0\|0\|0\|0\|0\|0 | 3\|0\|0 | 3 | 0.33 | 0.32 | 54\|46\|39 | 139 | 1.00 | 0.00 | 5.13 |
| CEP170 | *centrosomal protein 170* | 0\|0\|0\|3\|3\|0\|2 | 37\|31\|21 | 89 | 1.00 | 0.00 | 49\|38\|36 | 123 | 1.00 | 0.00 | 0.46 |
| LIMA1 | *LIM domain and actin binding 1* | 0\|0\|0\|0\|3\|0\|3 | 28\|25\|21 | 74 | 1.00 | 0.00 | 35\|33\|29 | 97 | 1.00 | 0.00 | 0.39 |
| MYL3/6/6B*** | *myosin light chain 3/6/6B* | 0\|0\|0 | 0\|0\|0 | 0 | | | 27\|27\|24 | 78 | 1.00 | 0.00 | 6.30 |
| PPP1CA | *protein phosphatase 1 catalytic subunit alpha* | 0\|0\|2\|0\|0\|0\|0\|0 | 14\|6\|0 | 20 | 0.58 | 0.11 | 25\|22\|21 | 68 | 1.00 | 0.00 | 1.72 |
| PCM1 | *pericentriolar material 1* | 0\|0\|0\|3\|0\|0\|0\|0 | 60\|31\|30 | 121 | 1.00 | 0.00 | 28\|19\|18 | 65 | 0.99 | 0.00 | -0.89 |
| EFHD1 | *EF-hand domain family member D1* | 0\|0\|0\|0\|0\|0\|0\|0 | 4\|0\|0 | 4 | 0.33 | 0.29 | 13\|12\|11 | 36 | 1.00 | 0.00 | 2.89 |
| TJP1 | *tight junction protein 1* | 0\|0\|0\|0\|0\|0\|0\|0 | 8\|8\|5 | 21 | 1.00 | 0.00 | 15\|9\|7 | 31 | 1.00 | 0.00 | 0.54 |
| SORBS1 | *sorbin and SH3 domain containing 1* | 0\|0\|0\|0\|0\|0\|0\|0 | 0\|0\|0 | 0 | | | 13\|6\|4 | 23 | 1.00 | 0.00 | 4.58 |
| SEPT2 | *septin 2* | 0\|0\|0\|0\|0\|0\|0\|0 | 0\|0\|0 | 0 | | | 10\|6\|5 | 21 | 1.00 | 0.00 | 4.46 |
| TP53BP2 | *tumor protein p53 binding protein 2* | 0\|0\|0\|0\|0\|0\|0\|0 | 7\|6\|2 | 15 | 0.97 | 0.00 | 7\|4\|0 | 11 | 0.67 | 0.08 | -0.42 |
| MAP7D3 | *MAP7 domain containing 3* | 0\|0\|0\|0\|0\|0\|0\|0 | 11\|11\|3 | 25 | 0.99 | 0.00 | 0\|0\|0 | 0 | | | -4.70 |

*iProphet>0.9; BFDR<0.01; unique peptides≥2; 8 controls collapsed to 2*

*\*FLNA and FLNB peptide counts have been merged due to their high homology (several identical tryptic peptides)*

*\*\* All peptides assigned to MYL12A are also found in MYL12B*

*\*\*\* most of the peptides assigned to MYL6 are also present in MYL3 sequence*

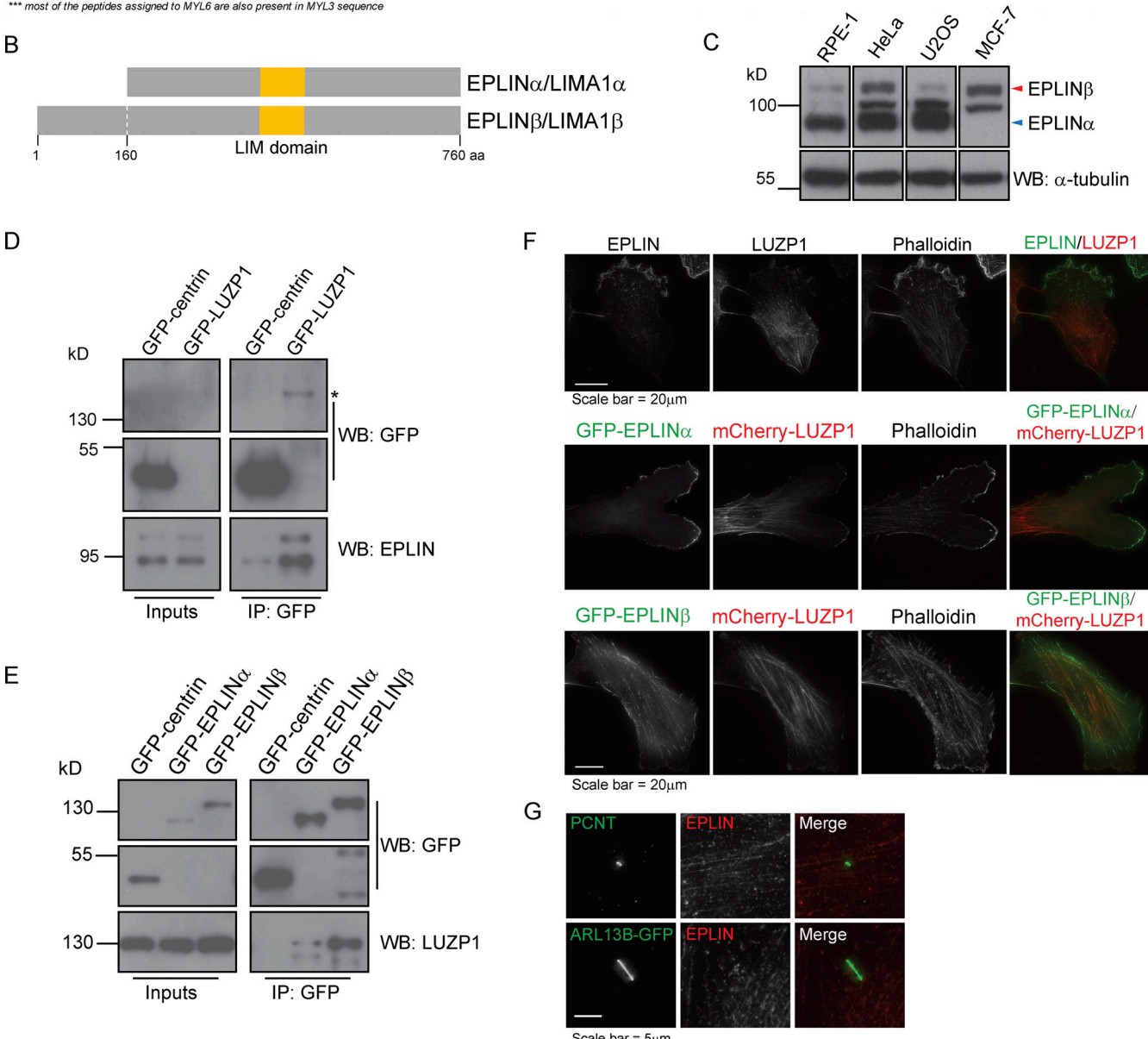

Figure 2. **EPLINα/β interacts with LUZP1. (A)** LUZP1 BioID interactors identified in cycling and serum-starved HEK293 cells. BFDR, Bayesian false discovery rate. **(B)** Schematic representation of human EPLIN isoforms α (600 aa) and β (760 aa), both of which contain a LIM domain. **(C)** EPLIN isoforms are expressed

differently in distinct cell lines. Western blot (WB) analysis of EPLIN expression in human RPE-1, HeLa, U2OS, and MCF-7 cells. **(D)** GFP-LUZP1 pulls down endogenous EPLIN. coIP experiments using protein extracts prepared from RPE-1 cells stably expressing GFP-centrin (control) or GFP-LUZP1. The fusion proteins were immunoprecipitated using GFP antibody–conjugated beads. GFP and EPLIN antibodies were used to detect the GFP fusions and endogenous EPLIN isoforms, respectively. Asterisk indicates the GFP-LUZP1 band. **(E)** Both EPLIN isoforms pull-down endogenous LUZP1. coIP experiments using protein extracts prepared from RPE-1 cells stably expressing GFP-centrin (control) or GFP-EPLINα/β. The fusion proteins were immunoprecipitated using GFP antibody-conjugated beads. GFP and LUZP1 antibodies were used to detect the GFP fusions and endogenous LUZP1, respectively. **(F)** EPLIN localizes to actin structures in RPE-1 cells. RPE-1 cells were stained with antibodies against EPLIN and LUZP1 (top panel). RPE-1 cells stably expressing GFP-EPLINα or β cells and mCherry-LUZP1 (middle and bottom panels) were stained with fluorophore-conjugated phalloidin. **(G)** EPLIN does not localize to the centrosome or the basal body. RPE-1 cells were stained for EPLIN and PCNT (centrosome marker; top panel). RPE-1 cells stably expressing ARL13B-GFP were stained for GFP and EPLIN.

---

controls. Interestingly, overexpressing both LUZP1 and EPLIN did not have an additive effect on ciliation prevention (Fig. 4 D). These results prompted us to investigate if depleting LUZP1 and EPLIN affects the levels of MyosinVa at the centrosome (Fig. 5, A and B). For this, WT RPE-1 cells were treated with a control siRNA or siRNAs targeting LUZP1 or EPLIN and were not serum starved. The cells were then stained for MyosinVa and its levels at the centrosome were quantified. This analysis revealed an increase in MyosinVa signal at the centrosome upon LUZP1 and EPLIN depletion (Fig. 5, A and B). This is consistent with these proteins having an actin-stabilizing role and their ability to rescue the effect of cytochalasin D on ciliation when overexpressed (Fig. 4 D). We also investigated if the expression of GFP-tagged LUZP1 and EPLIN had an effect on the levels of actin regulators and actin at the centrosome. Immunofluorescence (IF) analysis revealed significantly higher levels of the actin nucleator ARP2, part of the ARP2/3 complex, as well as actin at the centrosome in the overexpressing lines, compared with those of control GFP-only cells (Fig. S3, A–D). Although with the available data, we cannot rule out a noncentrosomal role for EPLIN and LUZP1, it appears plausible that increased actin nucleation at the centrosome by the ARP2/3 complex contributes, at least in part, to counteract the effect of cytochalasin D on ciliation.

Collectively, our results so far suggested that LUZP1 and EPLIN have similar functions regarding ciliogenesis regulation, possibly through their actin-associated roles. Yet, it remained unclear if they act in the same pathways. To test this, we used RNAi to simultaneously deplete LUZP1 and EPLIN in WT RPE-1 cells and assessed the effect of their depletion on cilia formation in the presence of serum. Our results showed that codepleting these proteins caused a significant increase in ciliation compared with the individual RNAi conditions (Fig. S3, E and F), suggesting LUZP1 and EPLIN may not fully overlap in the molecular mechanisms they are involved in. To further explore the role of both proteins, we tested the effect of cytochalasin D treatment combined with the RNAi of LUZP1 or EPLIN on ciliogenesis. For this, we used two control cell lines (WT and GFP-only RPE-1 cells) and the lines expressing GFP-tagged LUZP1, EPLINα, or EPLINβ (Fig. 5, C–F; and Fig. S3, G and H). The cells were transfected for 72 h in the presence of serum and treated with cytochalasin D or DMSO (vehicle) for the last 16 h. The control cell lines treated with DMSO showed the expected increase in ciliation caused by LUZP1 or EPLIN depletion (Fig. 5, C and F). Also as expected, treatment with cytochalasin D increased ciliation in the control cell lines transfected with non-targeting siRNAs. Strikingly, the same cell lines depleted of

LUZP1, but not EPLIN, showed an additional significant increase in the percentage of ciliated cells upon treatment with the drug (Fig. 5, C and F). One possible explanation for these results is that LUZP1 depletion destabilizes the actin cytoskeleton, sensitizing the cells to this actin-disrupting drug. This may be particularly relevant at the centrosome, where LUZP1 could have an important role in modulating actin dynamics during the centrosome to basal body transition. Alternatively, LUZP1 may have other functions that can explain its role in ciliogenesis outside of the centrosome, and what we observed is the additive effect of its depletion and the drug effect. Interestingly, silencing EPLIN did not produce the same effect in combination with cytochalasin D (Fig. 5, C and F), which may suggest it acts in the same pathway affected by the drug. In RPE-1 GFP-LUZP1 cells treated with DMSO and depleted of EPLIN, ciliation increased to a lesser extent than in the control cell lines in the same condition (Fig. 5, D and F). This suggests that overexpressing LUZP1 can rescue the effect of EPLIN depletion in cilia formation at least partially. In agreement with the aforementioned results (Fig. 4 D), treatment with cytochalasin D lead to a significantly lower level of induced ciliation in these cells. Also, EPLIN depletion and cytochalasin D treatment did not have a synergistic effect in increasing cilia formation (Fig. 5, D and F). Finally, in GFP-EPLINα– or GFP-EPLINβ–expressing cells treated with DMSO, LUZP1 depletion did not lead to a significant increase in ciliation, suggesting that the actin-stabilizing effect of overexpressed EPLIN rescues the LUZP1 RNAi phenotype (Fig. 5, E and F; and Fig. S3 H). Upon cytochalasin D treatment, EPLIN-overexpressing cells transfected with control siRNAs showed the expected modest increase in ciliogenesis. However, the same cells depleted of LUZP1 and treated with cytochalasin D showed ciliation levels similar to those of the control cell lines treated with the drug (Fig. 5, E and F; and Fig. S3 H). This suggests that EPLIN requires LUZP1 to counteract the actin disruption and ciliation increase caused by cytochalasin D. These results may well be related to the centrosomal localization of LUZP1 and the regulation of actin stability at this organelle. Indeed, centrosomal LUZP1 might be necessary to translate the actin stabilization effect of overexpressed EPLIN to the centrosome in extreme conditions of compromised actin functions, such as treatment with cytochalasin D. Further work will be necessary to determine the exact role of LUZP1, EPLIN, and their interactors (e.g. CDC14A) on actin-related processes at the centrosome or other cellular locales and how they specifically participate in the centrosome to basal body transition and primary cilia formation.

Here, we showed that human LUZP1 localizes to the centrosome, basal body, actin filaments, and midbody and associates

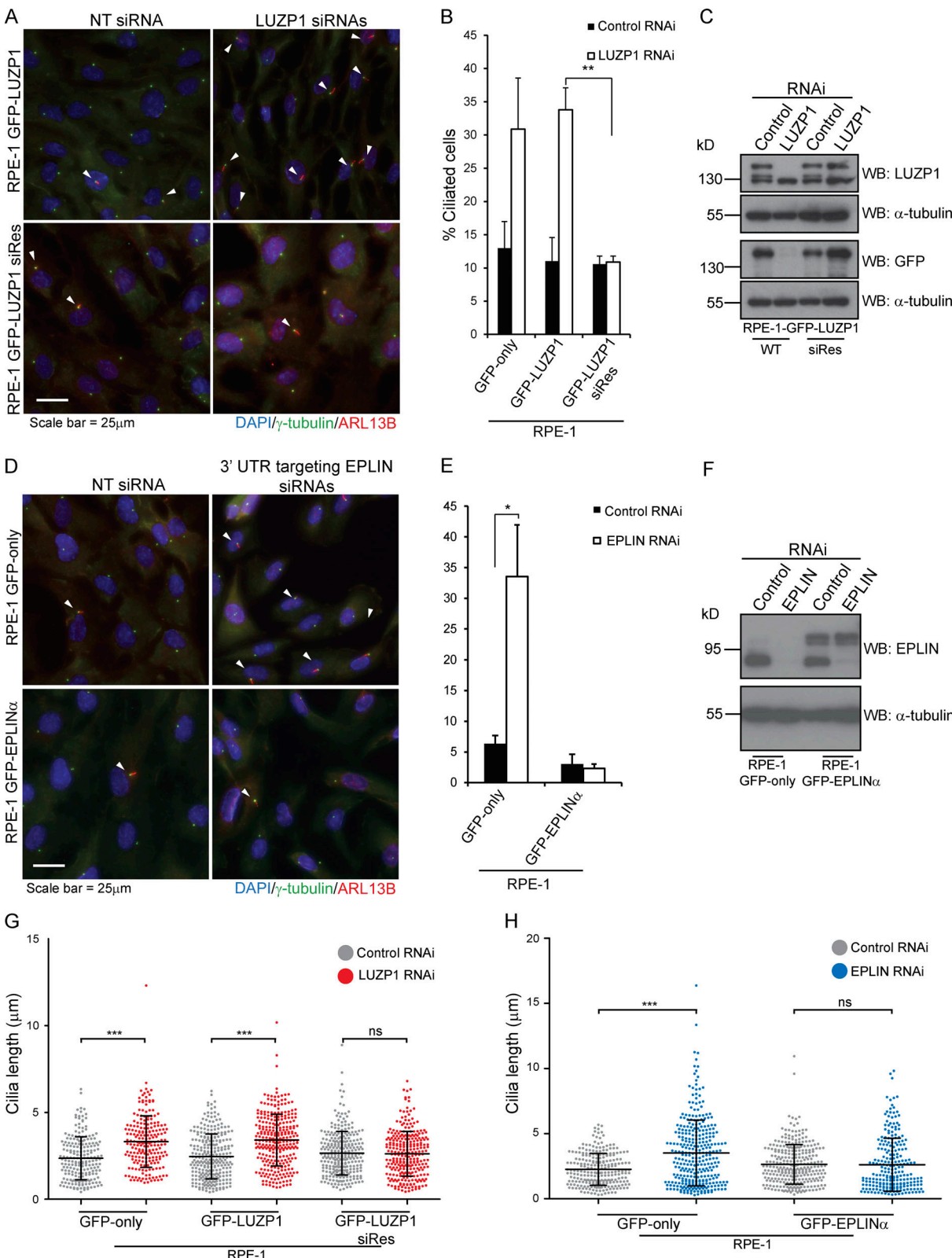

Figure 3. **LUZP1 and EPLIN are novel negative regulators of ciliogenesis. (A)** LUZP1 depletion increases ciliation in RPE-1 cells. IF analysis of RPE-1 cells stably expressing siRNA-sensitive or resistant GFP-LUZP1 and transfected with control (NTsiRNA) or siRNAs targeting LUZP1 for 72 h. The cells were stained for γ-tubulin (centrosome and basal body marker) and ARL13B (ciliary marker). DNA was stained with DAPI. The arrowheads indicate primary cilia. **(B)** Bar graph shows the mean percentage of ciliated cells (n > 200 cells per sample, three independent experiments) in both RPE-1 stable lines transfected with the indicated siRNAs for 72 h. Error bars indicate SD. **, P < 0.01 (Student's two-tailed t test). **(C)** Western blot analysis of endogenous and GFP-tagged LUZP1 in both stable lines transfected with control nontargeting (NT) or LUZP1-directed siRNAs for 72 h. **(D)** EPLIN silencing increases ciliation in RPE-1 cells. IF analysis

of RPE-1 cells stably expressing GFP-only or GFP-EPLINα and transfected with control (NT siRNA) or siRNAs targeting the 3′ UTR of EPLIN for 72 h. The cells were stained for γ-tubulin (centrosome and basal body marker) and ARL13B (ciliary marker). DNA was stained with DAPI. The arrowheads indicate primary cilia. **(E)** Bar graph shows the mean percentage of ciliated cells ($n > 200$ cells per sample, three independent experiments) in both stable lines transfected with the indicated siRNAs for 72 h. Error bars indicate SD. *, $P < 0.05$ (Student's two-tailed $t$ test). **(F)** Western blot analysis of endogenous and GFP-tagged EPLINα in the cells transfected with control nontargeting (NT), or LUZP1-directed siRNAs for 72 h. **(G and H)** LUZP1 and EPLIN depletion leads to increased ciliary length in RPE-1 cells. The graphs show the length of cilia measured in the indicated populations treated wither with control siRNA or siRNAs directed against LUZP1 or EPLIN. $n > 200$ cilia per condition (data pooled from three independent experiments). Bars correspond to the average and SD. ***, $P < 0.001$ (Student's two-tailed $t$ test); ns, nonsignificant.

mainly with centrosome/basal body components and actin-associated proteins like EPLIN. LUZP1 depletion in RPE-1 cells did not affect cell cycle progression. In fact, despite the midbody localization, no signs of cytokinesis failure (e.g., unresolved cellular bridges, multinucleated cells, or abnormal centrosome numbers) were observed in LUZP1-depleted cells. A role in cell division cannot, however, be ruled out due to potential residual levels of LUZP1. Silencing LUZP1 and EPLIN caused increased ciliation and ciliary lengthening. Critically, depleting these proteins increased the levels of MyosinVa at the centrosome, which also happens in the case of ciliation induction by cytochalasin D. Thus LUZP1 and EPLIN may play a role in the early steps of ciliogenesis. These results and the similarity to EPLIN in terms of loss-of-function phenotypes lead us to propose an actin-related role for LUZP1 supported by several lines of evidence. First, similar to EPLIN, LUZP1 overexpression stabilizes actin, as evidenced by brighter phalloidin staining of actin filaments. Also, overexpressing aa 1–496 caused a striking effect on actin organization consistent with a recent report showing that LUZP1 fragments (aa 1–500, and aa 400–500) cross-link actin in vitro (Wang and Nakamura, 2019). These results suggest that the C-terminal region of LUZP1 (aa 500–1,796) somehow masks/inhibits this actin-bundling domain. Second, the overexpression of LUZP1, as well as EPLIN, rescues the increased ciliation phenotype caused by cytochalasin D, possibly through, at least in part, increased levels of ARP2 and actin at the centrosome. Also, depleting LUZP1, but not EPLIN, sensitizes cells to cytochalasin D treatment, reflected by an additional increase in ciliation levels when the two treatments are combined. Some of these results have been validated in a recent report (Bozal-Basterra et al., 2019 Preprint).

The relevance of the interaction between LUZP1 and EPLIN remains unclear. There are similarities in loss-of-function and overexpression phenotypes for both proteins. Also, the overexpression of one can rescue, at least partially, the loss-of-function phenotype of the other. However, the codepletion of both genes leads to an additional increase in ciliation compared to individual knockdowns. Importantly, LUZP1 is required for EPLIN's role in counteracting the effect of cytochalasin D on ciliation. One key difference between the two proteins relates to their localization. LUZP1, but not EPLIN, localizes to the centrosome/basal body, where it may regulate actin dynamics. This could be critical to translate EPLIN's effect on actin to the site of cilia formation in situations of a compromised cytoskeleton, as under cytochalasin D treatment. Alternatively, LUZP1 may have other roles not necessarily linked to actin. Finally, LUZP1 and EPLIN seem to colocalize to a certain degree along actin filaments but accumulate at distinct actin-associated structures. It is therefore possible that while these proteins may work together in the regulation of some cellular processes, they may well possess independent roles in others, which warrants further investigation.

Terminal deletions of chromosome 1p36, with a consequent loss of LUZP1, cause a developmental disease (1p36 Deletion Syndrome), with potential clinical manifestations being short stature, affected brain development (e.g. microcephaly), hearing loss, congenital heart disease, and renal disease (Zaveri et al., 2014). The phenotypes of a LUZP1-knockout mouse model (cardiovascular defects and cranial neural tube closure failure) agree with the hypothesis that the loss of the human gene contributes to this disorder. Importantly, the mice show ectopic sonic-hedgehog signaling, which may be due to perturbed ciliation (Hsu et al., 2008). On the other hand, EPLIN loss of expression has been associated with cancer. This has been shown to affect cancer cell adhesion and migration and increase metastatic potential (Collins et al., 2018; Jiang et al., 2008; Liu et al., 2012; Sanders et al., 2010; Zhang et al., 2011). Like LUZP1, EPLIN had never been implicated in cilia biology. However, the roles of these proteins as negative regulators of ciliogenesis likely impact the multiple cilia-dependent signaling pathways contributing to the etiology of their associated diseases. Indeed, several of 1p36 deletion syndrome phenotypes are similar to those of well-established ciliopathies, and perturbed ciliary signaling contributes to the development of certain cancers.

## Materials and methods
### Cell culture
Flp-In T-REx HEK293 cells were grown in DMEM supplemented with 10% FBS, GlutaMAX, zeocin (100 μg/ml), and blasticidin (3 μg/ml). Flp-In T-REx 293 stable lines expressing Flag-BirA* or Flag-BirA*-LUZP1 were maintained as above, with the addition of hygromycin (200 μg/ml) or puromycin (1 μg/ml). HeLa and MCF-7 cells were grown in DMEM supplemented with 10% FBS and GlutaMAX. hTERT-RPE-1 cells (WT and stable lines) were grown in DMEM/F12 supplemented with 10% FBS, GlutaMAX and sodium bicarbonate (1.2 g/liter). U-2 OS cells were grown in McCoy 5A medium supplemented with 10% FBS and GlutaMAX. All cells were cultured in a 5% $CO_2$ humidified atmosphere at 37°C. hTERT-RPE-1 cells stably expressing GFP-centrin were a generous gift from Dr. A. Khodjakov (Wadsworth Center, New York State Department of Health, Albany, NY).

### Molecular cloning
The human LUZP1 (NM_001142546.1) and EPLIN (NP_001107018.1; NM_001113547.2) coding sequences were amplified from a testes

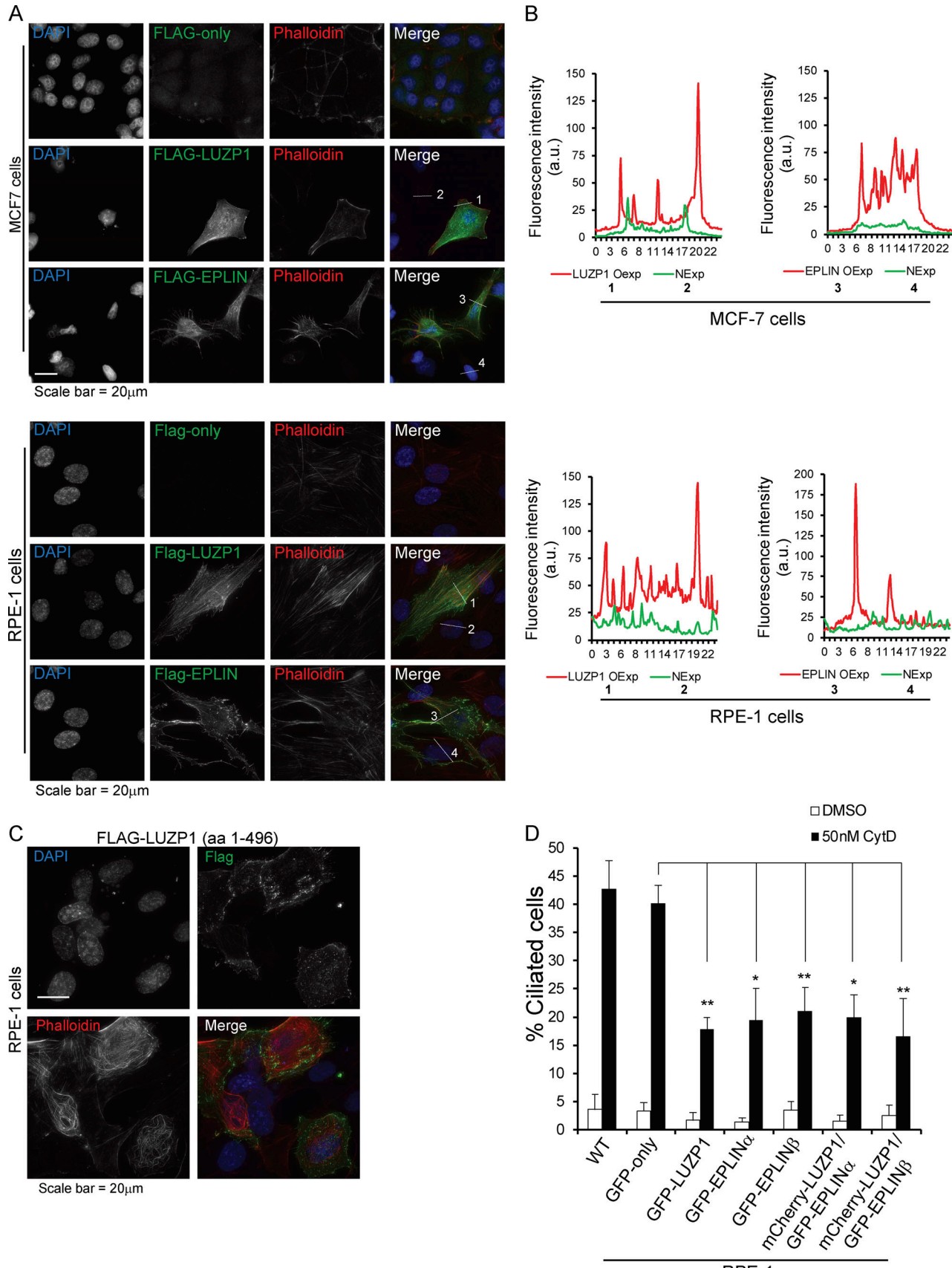

Figure 4.  **LUZP1 and EPLIN have actin stabilization roles. (A)** IF analysis of MCF-7 (top panel) and RPE-1 (bottom panel) cells transiently transfected with a control empty FLAG vector or plasmids to express FLAG-LUZP1 or FLAG-EPLINβ. The fusion proteins were detected with a FLAG antibody. The actin

cytoskeleton was stained with fluorophore-conjugated phalloidin, and DNA was stained with DAPI. **(B)** The graphs show fluorescence intensity of phalloidin in MCF-7 and RPE-1 cells overexpressing LUZP1 or EPLIN and neighboring cells measured along the lines indicated in A. NExp, nonexpressing; OExp, over-expressing. **(C)** IF analysis of RPE-1 cells transiently transfected with a FLAG-LUZP1 (aa 1–496) construct. The actin cytoskeleton was stained with fluorophore-conjugated phalloidin, and DNA was stained with DAPI. **(D)** Bar graph shows the mean percentage of ciliated cells ($n > 200$ cells per sample, three independent experiments) in the indicated cell lines treated with DMSO or 50 nM cytochalasin D (CytD). Error bars indicate SD. *, $P < 0.05$; **, $P < 0.01$ (Student's two-tailed $t$ test).

cDNA sample using primers (Table S2) containing appropriate restriction sites. The PCR products were digested and ligated into the following vectors: pCMV-TO/FRT-Emerald (GFP), pcDNA5-TO/FRT-mCherry (only LUZP1), pcDNA5-TO/FRT-FLAG, and pPUR 5′/FRT/TO-FLAG-BirA* (only LUZP1). The GFP-LUZP1, GFP-EPLIN, and mCherry-LUZP1 fusion coding sequences were amplified by PCR and subcloned into the lentiviral vector pHR-SIN-SFFV. The LUZP1 truncation mutants were generated by amplifying the respective coding sequences by PCR, followed by ligation into the pCMV-TO/FRT-Emerald (GFP) or pcDNA5-TO/FRT-FLAG vectors.

### Lentiviral production and generation of the hTERT RPE-1 stable cell lines

For the production of lentiviral particles, HEK293T cells were cotransfected with pHR-SIN-SFFV-GFP-LUZP1, pHR-SIN-SFFV-mCherry-LUZP1, pHR-SIN-SFFV-GFP-EPLINα, or pHR-SIN-SFFV-GFP-EPLINβ and the second-generation packaging (pCMV-dR8.74psPAX2) and envelope (pMD2.G) plasmids using the Lipofectamine 3000 transfection reagent (Invitrogen) according to the manufacturer's instructions. Lentiviral particles in conditioned media from HEK 293T cells, collected at 48 h after transfection, were used to transduce hTERT RPE-1 cells. GFP- or GFP + mCherry–positive cells were cell sorted to establish the final fluorescent cell lines.

### Treatment with cytoskeleton drugs

To depolymerize MTs, hTERT RPE-1 GFP-LUZP1 cells were incubated with 30 µM nocodazole for 1 h at 37°C before they were fixed and processed for IF.

To perturb the actin cytoskeleton, cells were treated with 50 nM cytochalasin D for 16 h. In the case of RNAi experiments, the cytochalasin treatment happened during the last 16 h of the 72 h of gene silencing. For all experiments, the cells were treated with the same volume of DMSO as a control.

### Generation and characterization of stable and inducible HEK293 pools for BioID

Flp-In T-REx HEK293 cells were cotransfected with pOG44 (Flp-recombinase expression vector) and the pPUR 5′/FRT/TO-FLAG-BirA*-LUZP1. Transfections were performed with Lipofectamine 2000 according to manufacturer's instructions. After transfection, cells were selected with 1 µg/ml puromycin. Cells were incubated in one of three conditions for 24 h: (1) 1 µg/ml tetracycline (BioShop) and 50 µM biotin (Sigma-Aldrich), (2) 1 µg/ml tetracycline only, or (3) no tetracycline or biotin added. The cells were fixed with ice-cold methanol and processed for IF. Biotinylated proteins were detected using fluorophore-conjugated streptavidin (Invitrogen).

### BioID sample preparation

BioID (Roux et al., 2012) was performed essentially as described previously (Coyaud et al., 2015). Independent replicates of five 15-cm plates of subconfluent (60%) stably expressing FLAG-BirA* alone (eight replicates) or FLAG-BirA*-LUZP1 (two replicates) were incubated for 24 h in complete media (and in serum-free media for 48 h before biotin addition for the ciliated condition) supplemented with 1 µg/ml tetracycline (Sigma-Aldrich) and 50 µM biotin (BioShop). Cells were collected and pelleted (2,000 rpm, 3 min), the pellet was washed twice with PBS, and dried pellets were snap frozen. The cell pellet was resuspended in 10 ml lysis buffer (50 mM Tris-HCl, pH 7.5, 150 mM NaCl, 1 mM EDTA, 1 mM EGTA, 1% Triton X-100, 0.1% SDS, 1:500 protease inhibitor cocktail [Sigma-Aldrich], and 1:1,000 benzonase nuclease [Novagen]) and incubated on an end-over-end rotator at 4°C for 1 h, briefly sonicated to disrupt any visible aggregates, and then centrifuged at 45,000 ×g for 30 min at 4°C. Supernatant was transferred to a fresh 15-ml conical tube. 30 µl packed, preequilibrated Streptavidin Sepharose beads (GE) were added and the mixture incubated for 3 h at 4°C with end-over-end rotation. Beads were pelleted by centrifugation at 2,000 rpm for 2 min and transferred with 1 ml lysis buffer to a fresh Eppendorf tube. Beads were washed once with 1 ml lysis buffer and twice with 1 ml of 50 mM ammonium bicarbonate (pH 8.3). Beads were transferred in ammonium bicarbonate to a fresh centrifuge tube and washed two more times with 1 ml ammonium bicarbonate buffer. Tryptic digestion was performed by incubating the beads with 1 µg MS-grade TPCK trypsin (Promega) dissolved in 200 µl of 50 mM ammonium bicarbonate (pH 8.3) overnight at 37°C. The following morning, 0.5 µg MS-grade TPCK trypsin was added, and beads were incubated two additional hours at 37°C. Beads were pelleted by centrifugation at 2,000 ×g for 2 min, and the supernatant was transferred to a fresh Eppendorf tube. Beads were washed twice with 150 µl of 50 mM ammonium bicarbonate, and these washes were pooled with the first eluate. The sample was lyophilized and resuspended in buffer A (0.1% formic acid). One fifth of the sample was analyzed per MS run.

### MS

Analytical columns (75 µm inner diameter) and precolumns (150 µm inner diameter) were made in-house from fused silica capillary tubing from InnovaQuartz and packed with 100 Å C18–coated silica particles (Magic; Michrom Bioresources). Peptides were subjected to liquid chromatography electrospray ionization tandem MS using a 120-min reversed-phase (100% water–100% acetonitrile, 0.1% formic acid) buffer gradient running at 250 nl/min on a Proxeon EASY-nLC pump in-line with a hybrid Linear Trap Quadrupole-Orbitrap Velos mass spectrometer (Thermo Fisher Scientific). A parent ion scan was performed in the Orbitrap using a resolving power of 60,000, and then ≤20 of the most intense peaks were selected for MS/MS (minimum ion count of 1,000 for activation) using standard

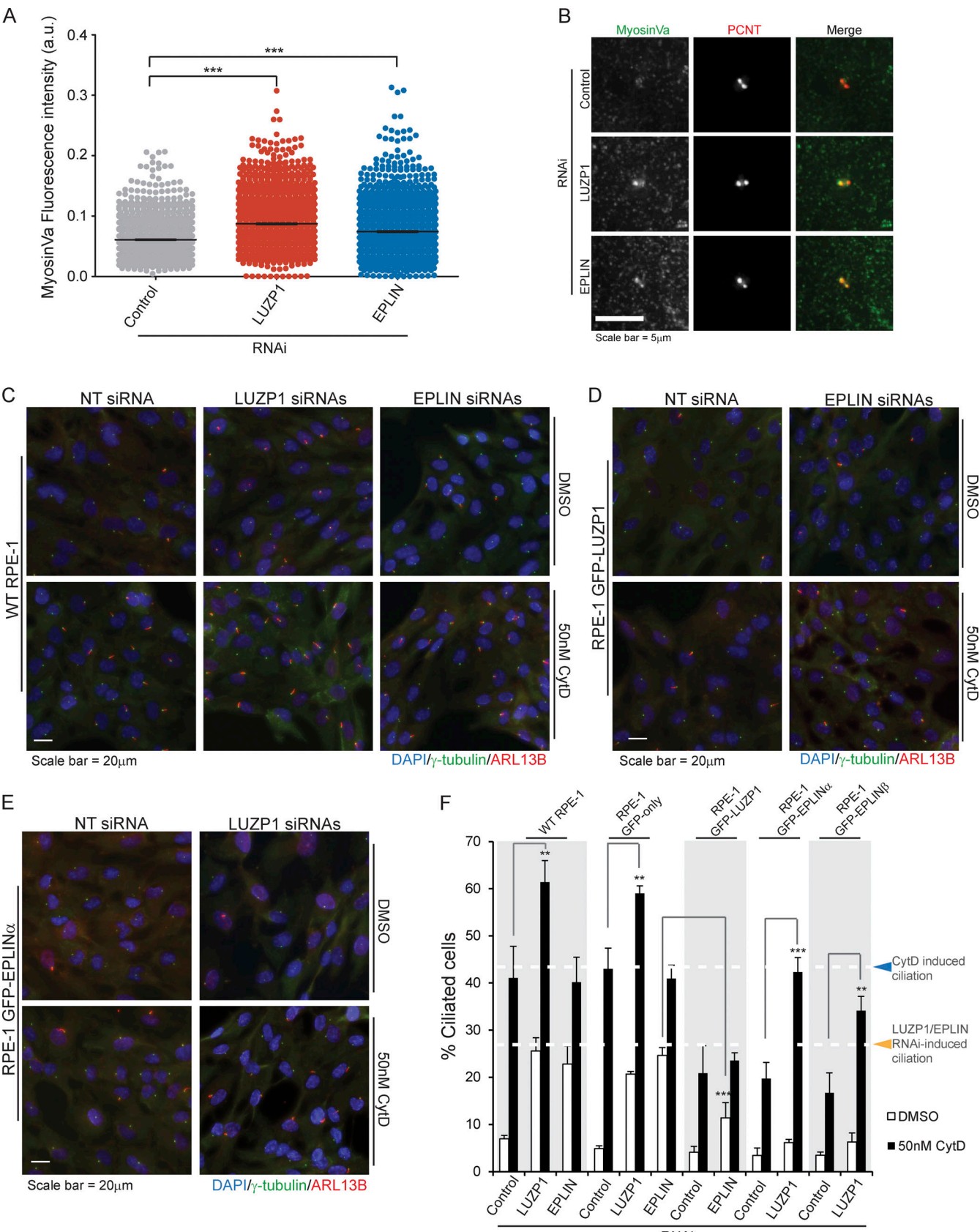

Figure 5. **EPLIN requires LUZP1 to rescue cytochalasin D–induced ciliation. (A)** MyosinVa fluorescence intensity at the centrosome of RPE-1 cells transfected with control siRNAs or siRNAs directed to LUZP1 or EPLIN. The graph shows data pooled from three independent experiments. The bar indicates average intensity. \*\*\*, P < 0.01 (Student's two-tailed *t* test). **(B)** Representative images of control and LUZP1 and EPLIN depleted RPE-1 cells stained for

MyosinVa and PCNT (centrosome marker). **(C–E)** WT and transgenic RPE-1 cells were transfected for 72 h with control siRNA or siRNAs targeting LUZP or EPLIN, as indicated. During the last 16 h of transfection, cells were treated either with DMSO (control) or 50 nM cytochalasin D. Cells were stained for γ-tubulin (centrosome and basal body marker) and ARL13B (ciliary marker). DNA was stained with DAPI. **(F)** Bar graph shows the mean percentage of ciliated cells (n > 200 cells per sample, three independent experiments) in the indicated cell lines transfected with control, LUZPI, or EPLIN siRNAs and treated with DMSO or 50 nM cytochalasin D. Error bars indicate SD. **, P < 0.01; ***, P < 0.001 (Student's two-tailed t test).

collision induced dissociation fragmentation. Fragment ions were detected in the Linear Trap Quadropole. Dynamic exclusion was activated such that MS/MS of the same m/z (within a range of 15 ppm; exclusion list size, 500) detected twice within 15 s were excluded from analysis for 30 s. For protein identification, raw files were converted to the .mzXML format using Proteowizard (Kessner et al., 2008) and then searched using X! Tandem (Craig and Beavis, 2004) and COMET (Eng et al., 2013) against the human RefSeq version 45 database (containing 36,113 entries). Data were analyzed using the transproteomic pipeline (Deutsch et al., 2010; Pedrioli, 2010) via the ProHits software suite (v3.3; Liu et al., 2010). Search parameters specified a parent ion mass tolerance of 15 ppm and an MS/MS fragment ion tolerance of 0.4 D, with up to two missed cleavages allowed for trypsin. Variable modifications of +16@M and W, +32@M and W, +42@N-terminus, and +1@N and Q were allowed. Proteins identified with an iProphet cutoff of 0.9 (corresponding to ≤1% false discovery rate) and at least two unique peptides were analyzed with SAINT (Significance Analysis of Interactome) Express v.3.3.1, a probabilistic method for scoring bait–prey interactions against negative controls. The SAINT algorithm uses spectral counts or protein intensities as the input to calculate the probability of true interaction, allowing for the selection of high-confidence interactions (Teo et al., 2016). Eight control runs (from cells expressing the Flag-BirA* epitope tag) were collapsed to the two highest spectral counts for each prey and compared with the two biological replicates of *LUZP1* BioID (two technical replicates of each; four runs collapsed to three highest spectral counts for each prey) in normal and serum-starved conditions. High-confidence interactors were defined as those with Bayesian false discovery rate ≤0.01.

### RNAi

To silence LUZP1 and EPLIN, RPE-1 cells ($10^5$ cells seeded in 6-well plates) were transfected with 40 nM (final concentration) of a pool of two siRNAs targeting either gene obtained from Dharmacon (ON-TARGET plus) using the Lipofectamine RNAiMAX transfection reagent (Invitrogen) according to the manufacturer's instructions. The Luciferase GL2 Duplex nontargeting siRNA from Dharmacon was used as a negative control. Gene silencing was performed for 72 h. The siRNA sequences are listed in Table S2. When indicated, the cells were serum starved (DMEM/F12 supplemented with GlutaMAX and sodium bicarbonate [1.2 g/liter] but without serum) for 72 h to induce the formation of primary cilia.

### RNAi rescue experiments

hTERT RPE-1 GFP-only and hTERT RPE-1 GFP-LUZP1 stable lines expressing the siRNA resistant or sensitive LUZP1 transgenes were transfected with 40 nM (final concentration) of a pool of two siRNAs (Dharmacon ON-TARGET plus) directed against

LUZP1 using the Lipofectamine RNAiMAX transfection reagent (Invitrogen) according to the manufacturer's instructions. The Luciferase GL2 Duplex nontargeting siRNA from Dharmacon was used as a negative control. Gene silencing was performed for 72 h. hTERT RPE-1 GFP-only and hTERT RPE-1 GFP-EPLINα stable lines were transfected with 40 nM (final concentration) of a pool of two siRNAs (Dharmacon ON-TARGET plus) directed against the 3′ UTR of EPLIN transcripts. Gene silencing was performed for 72 h.

### Fluorescence microscopy

For IF, the cells were fixed with cold methanol (10 min at –20°C) or 4% paraformaldehyde at room temperature for 10 min. The cells were then blocked with 0.2% Fish Skin Gelatin (Sigma-Aldrich) in 1× PBS (20 min) or with blocking buffer (PBS containing 1% BSA and 0.5% Triton X-100), incubated with the primary antibodies in blocking solution (1 h), washed with blocking solution, and incubated with fluorophore-conjugated secondary antibodies (Molecular Probes) and DAPI in blocking solution (1 h). After a final wash in blocking solution, the coverslips were mounted on glass slides by inverting them onto mounting solution (ProLong Gold antifade; Molecular Probes). For the characterization of the HEK293 Flag-BirA*-LUZP1 line, the cells were also incubated with fluorophore-conjugated streptavidin (Molecular Probes). The cells were imaged on a DeltaVision (Applied Precision) imaging system equipped with an IX71 microscope (Olympus), charge-coupled device camera (CoolSNAP HQ2 1,024 × 1,024; Roper Scientific) and a 60×/1.42 NA Plan-Apochromat oil-immersion objective (Olympus). Z stacks (0.2 µm apart) were collected, deconvolved using softWoRx (v5.0, Applied Precision), and are shown as maximum intensity projections (pixel size, 0.1064 µm). Primary and secondary antibodies used are listed in Table S2.

For live imaging, the cells were seeded in Nunc Lab-Tek Chamber Slides and imaged on the DeltaVision system with temperature and $CO_2$ control, using a 40×/1.35 NA oil-immersion objective (Olympus). Time lapse was 5 min. Z stacks (1 µm apart) were collected and deconvolved using softWoRx (v5.0, Applied Precision) and are shown as maximum intensity projections (pixel size, 0.33463 µm). Images were analyzed with FIJI (ImageJ; National Institutes of Health; cilia length measurements and line profile fluorescence measurements) or CellProfiler (MyosinVa fluorescence intensity measurements) software.

For confocal imaging, the images were acquired using Nikon Ti2E/AIR-HD25 confocal microscope using 60× oil-immersion Plan-Apochromat lambda objective and are shown as maximum intensity projection. Confocal images were analyzed using the IMARIS software suite (Bitplane) for the quantitation of fluorescence intensity. Three-dimensional images were

reconstructed using z-stacks in the IMARIS software suite. For performing intensity quantification, a sphere of 2 µm diameter (average size of the pole is 2 µm; Colello et al., 2012; Lecland and Lüders, 2014) was drawn around the pericentrin to calculate the integrated fluorescence intensity of actin and ARP2 around the centrosome. The total integrated intensity graphs were drawn using GraphPad Prism software.

### Cell proliferation assay

Time-lapse imaging of siRNA-transfected cells was done using an IncuCyte S3 Live Cell Analysis System (Essen BioScience) using a 10× objective. Phase images were acquired every 3 h for 72 h in 5% $CO_2$ humidified atmosphere at 37°C. Cell proliferation was quantified using the IncuCyte S3 Cell by Cell analysis module, which measures cell confluence during the time-lapse experiment.

### Western blotting

For Western blots, the cells were collected, lysed in Laemmli buffer, and treated with benzonase nuclease (Sigma-Aldrich). Proteins were separated by loading whole-cell lysates onto an 8–10% SDS-PAGE gel for electrophoresis and then transferred to a polyvinylidene fluoride membrane (Immobilon-P; Millipore). Membranes were incubated with primary antibodies in TBST (TBS, 0.1% Tween-20) in 5% skim milk powder (BioShop), supplemented with 2.5% BSA Fraction V (OmniPur) in the case of FLAG Western blots. Blots were washed three times 10 min in TBST and then incubated with secondary HRP-conjugated antibodies. Western blots were developed using SuperSignal reagents from Thermo Fisher Scientific.

### coIP followed by Western blot

For coIP of Flag fusions, the respective HEK293 stable lines were incubated with tetracycline (1 µg/ml) for 24 h, after which they were washed with 1x PBS, harvested, and frozen at –80°C or lysed immediately (50 mM Hepes, pH 8, 100 mM KCl, 2 mM EDTA, 10% glycerol, 0.1% NP-40, 1 mM DTT, and protease inhibitors) for 30 min on ice. The lysates were then frozen in dry ice for 5 min and then thawed and centrifuged for 20 min at 16,000 ×$g$ at 4°C. The cleared lysates were then incubated with ANTI-FLAG M2 Affinity Gel (Sigma-Aldrich) overnight at 4°C. A fraction of the protein extracts (inputs) were saved before the incubation with the beads. After incubation, beads were pelleted and washed with lysis buffer. Samples (inputs and IPs) were prepared for SDS-PAGE by addition of Laemmli buffer and boiling. Proteins were transferred to polyvinylidene fluoride membranes (Immobilon-P; Millipore) and probed with antibodies to detect the FLAG fusions and endogenous proteins. The GFP coIPs were performed similarly using extracts from HEK293 or RPE-1 cells expression GFP-fusions. Protein G Sepharose 4 Fast Flow beads (P3296; Sigma-Aldrich) were incubated with 2 µg GFP antibody raised in Goat for 2 h at 4°C and then washed with lysis buffer. GFP antibody-conjugated beads were then used to pull down the GFP fusions. Primary and secondary antibodies used are listed in Table S2.

### Statistical methods

All P values are from two-tailed unpaired Student $t$ tests. All error bars represent SD. Individual P values, experiment sample numbers, and the number of replicates used for statistical testing are reported in corresponding figure legends (***, P < 0.001; **, P < 0.01; *, P < 0.05).

### Online supplemental material

Fig. S1 shows IF subcellular localization analyses of LUZP1 and EPLIN fusion proteins in human cell lines. Fig. S2 shows coIP experimental data validating the LUZP1–EPLIN interaction in HEK293 cells. It also shows the effect of LUZP1 depletion on ciliation in serum-starved RPE-1 cells, the effect LUZP1 or EPLIN overexpression on ciliation in RPE-1 cells, and the effect of LUZP1 and EPLIN depletion on cell proliferation in RPE-1 cells. Fig. S3 shows IF analyses and respective quantifications of ARP2 and actin levels in LUZP1 and EPLIN overexpressing cells. The figure also shows the quantification of ciliation in cells depleted of LUZP1 and/or EPLIN and the respective Western blot data showing the depletion of the respective proteins. Finally, this figure shows the Western blot results demonstrating depletion of LUZP1 or EPLIN in RPE-1 stable lines in the cytochalasin D experiments as well as the ciliation IF analyses in RPE-1 cells expressing GFP-EPLINβ. Video 1 shows the dynamics of GFP-LUZP1 localizations during the cell cycle in RPE-1 cells. Table S1 contains the BioID-MS data. Table S2 lists the primers, siRNAs, and antibodies used in this work.

## Acknowledgments

We are grateful to Dr. Johnny Tkach, Dr. Ladan Gheiratmand, and Dr. Mikhail Bashkurov for very fruitful discussions during the development of this work.

J. Gonçalves was partially funded by Fundação para a Ciência e a Tecnologia (postdoctoral fellowship SFRH/BPD/75847/2011). This work was funded by operating grants from the Canadian Institutes of Health Research (MOP#142492 and FDN#167279) and the Krembil Foundation to L. Pelletier.

The authors declare no competing financial interests.

Author contributions: J. Gonçalves conceived the project, designed the research plan, and performed the experiments and data analyses. A. Sharma analyzed the centrosomal levels ARP2 and actin in RPE-1 stable lines. Sample preparation for MS was carried out by E.M.N. Laurent, E. Coyaud, and B. Raught performed the MS and the related analyses. J. Gonçalves wrote the manuscript with contributions from the other authors. L. Pelletier supervised and funded the project.

Submitted: 15 August 2019

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

# Supplemental material

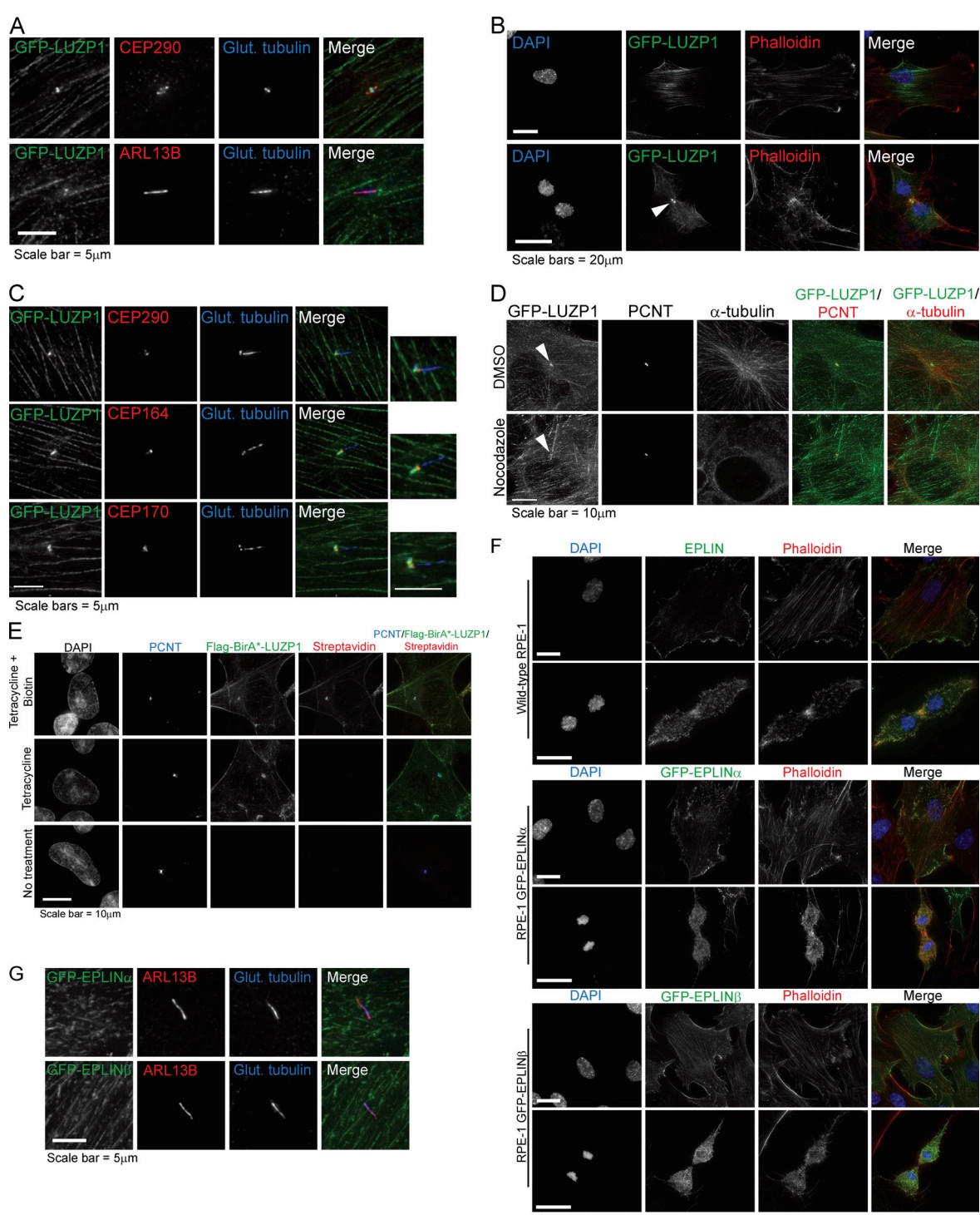

Figure S1.    **LUZP1 and EPLIN subcellular localization analyses. (A)** LUZP1 localizes to the centrosome and basal body. RPE-1 cells stably expressing GFP-LUZP1 were stained for GFP, glutamylated tubulin (centrosome and ciliary marker), ARL13B (cilia marker), and CEP290 (centriolar satellite). **(B)** LUZP1 localizes to actin filaments and the midbody (indicated by arrowhead). RPE-1 cells stably expressing GFP-LUZP1 were stained for GFP. The actin cytoskeleton was stained with fluorophore-conjugated phalloidin, and DNA was stained with DAPI. **(C)** LUZP1 localizes to the proximal domain of the centriole/basal body. RPE-1 cells stably expressing GFP-LUZP1 were stained for GFP, glutamylated tubulin (centrosome and ciliary marker), and CEP290 (centrosome and transition zone marker; top panel), CEP164 (distal appendage/transition fiber marker), or CEP170 (subdistal appendage marker). **(D)** IF analysis of RPE-1 GFP-LUZP1 stable cells treated with DMSO or nocodazole. Cells were stained for GFP, PCNT, and α-tubulin. Arrowheads indicate the centrosome. **(E)** IF analysis of HEK293 FLAG-BirA*-LUZP1 stable/inducible cells without treatment, treated with tetracycline only, or treated with tetracycline and biotin. Cells were stained with antibodies against FLAG and PCNT. Biotinylated proteins were detected with fluorophore-conjugated streptavidin and DNA with DAPI. **(F)** IF analysis of EPLIN subcellular localization. RPE-1 cells were stained with an antibody against EPLIN and phalloidin (top panel). RPE-1 cells stably expressing GFP-EPLINα or GFP-EPLINβ were stained with an antibody against GFP and phalloidin (middle and bottom panels). DNA was stained with DAPI. **(G)** RPE-1 cells stably expressing GFP-EPLINα or GFP-EPLINβ were stained with antibodies against GFP, ARL13B (cilia marker), and glutamylated tubulin (centriole and cilia marker).

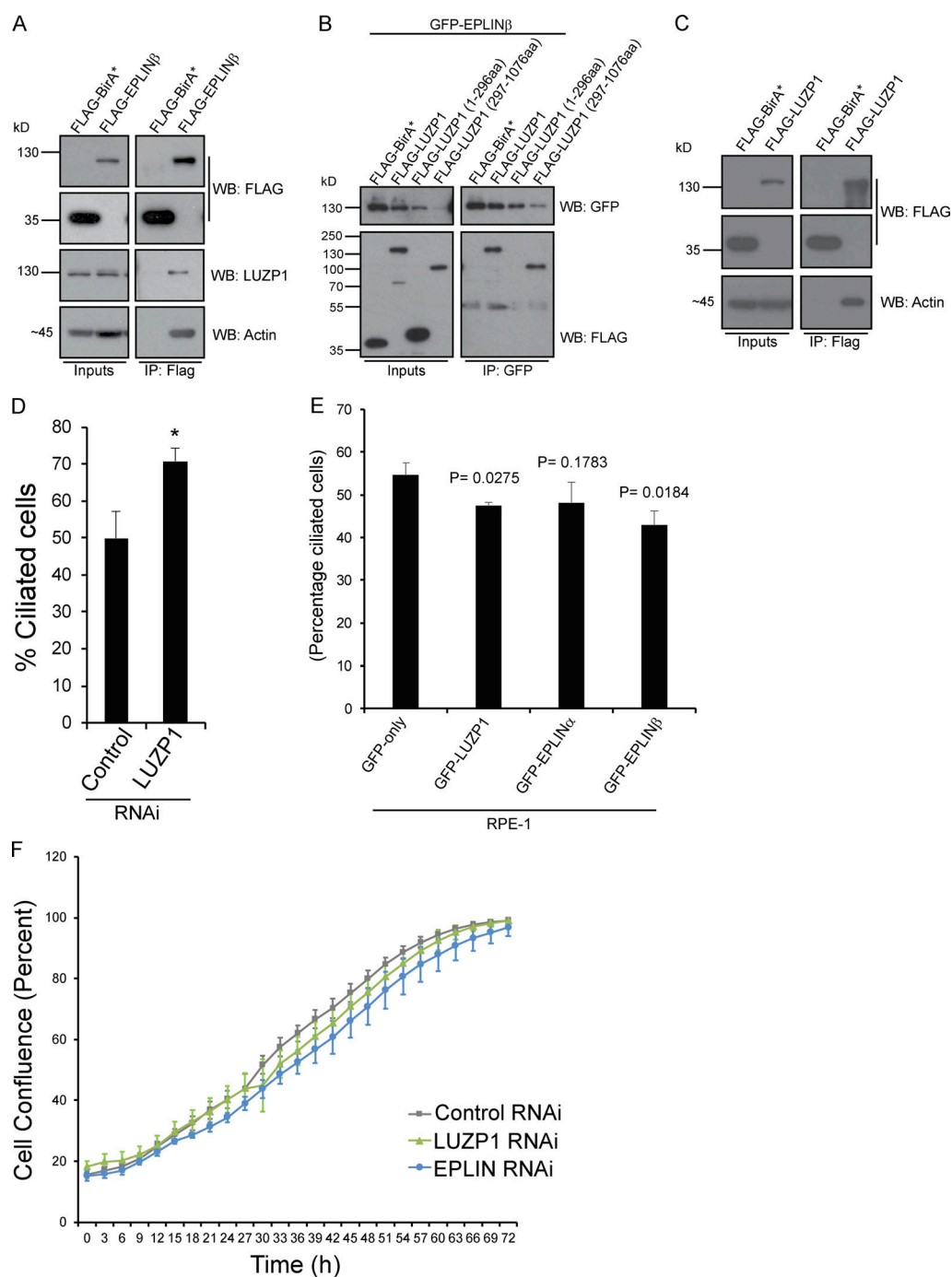

Figure S2. **LUZP1 and EPLIN interaction and functional studies. (A)** FLAG-EPLINβ pulls down endogenous LUZP1. coIP experiments using protein extracts prepared from HEK293 cells stably expressing FLAG-BirA* (control) or FLAG-EPLINβ. The fusion proteins were immunoprecipitated using FLAG antibody-conjugated beads. FLAG, LUZP1, and actin antibodies were used to detect the FLAG fusions, endogenous LUZP1, and actin, respectively. **(B)** GFP-EPLINβ pulls down the LUZP1 truncation mutant lacking the N-terminal domains. coIP experiments using protein extracts prepared from HEK293T stable cells expressing FLAG-BirA* (control), FLAG-LUZP1, FLAG-LUZP1 (1-296aa) or FLAG-LUZP1 (297–1,076 aa), and GFP-EPLINβ. The GFP-fusion was immunoprecipitated using GFP antibody-conjugated beads. GFP and FLAG antibodies were used to detect the GFP and FLAG fusions, respectively. **(C)** FLAG-LUZP1 pulls-down actin. coIP experiments using protein extracts prepared from HEK293 cells stably expressing FLAG-BirA* (control) or FLAG-LUZP1. The fusion proteins were immunoprecipitated using FLAG antibody-conjugated beads. FLAG and actin antibodies were used to detect the FLAG fusions and endogenous actin, respectively. **(D)** LUZP1 depletion increases ciliation in RPE-1 cells. WT RPE-1 cells were transfected with control (NTsiRNA) or siRNAs targeting LUZP1and serum-starved for 72 h. Bar graph shows the mean percentage of ciliated cells (n > 200 cells per sample, three independent experiments) in RPE-1 cells transfected with the indicated siRNAs for 72 h. Error bars indicate SD. *, P < 0.05 (Student's two-tailed t test). **(E)** Overexpressing LUZP1 or EPLIN causes a mild decrease in ciliation in RPE-1 cells. RPE-1 cells stably expressing GFP-only, GFP-LUZP1, GFP-EPLINα, or GFP-EPLINβ were serum starved for 72 h, after which the percentage of ciliated cells was determined. The graph shows data from three independent experiments. P values were determined by Student's two-tailed t test. **(F)** LUZP1 and EPLIN silencing does not affect cell proliferation. RPE-1 cells were transfected with a control siRNA or siRNAs targeting LUZP1 or EPLIN. Cell confluence was assessed by time-lapse imaging (determined by phase-contrast images). Error bars represent SD.

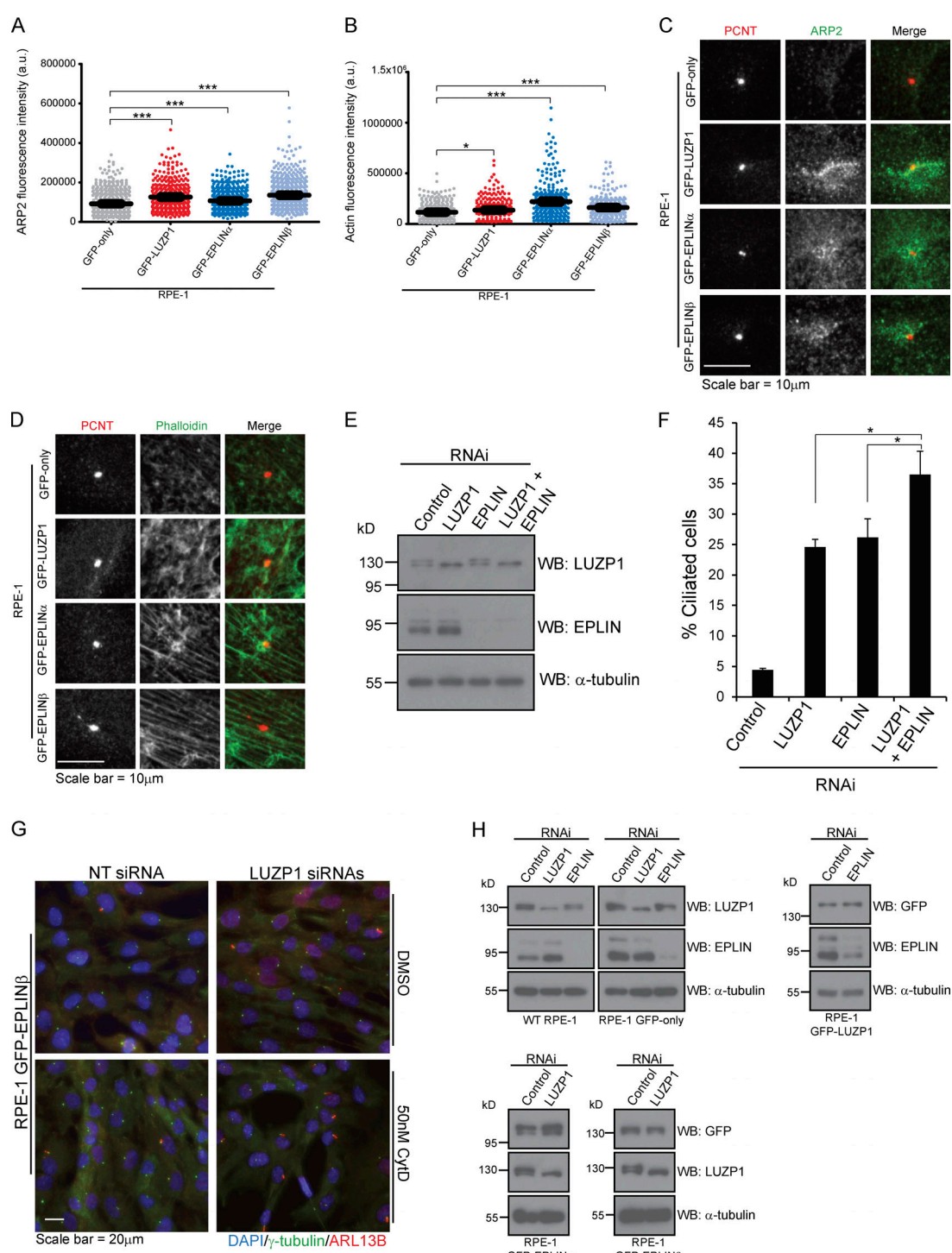

Figure S3. **LUZP1 is an actin-stabilizing protein. (A and B)** ARP2 and actin fluorescence intensity at the centrosome of RPE-1 cells stably expressing GFP-only, GFP-LUZP1, GFP-EPLINα, or GFP-EPLINβ. The bar indicates total integrated intensity. The bar indicates average intensity. ***, P < 0.01 (Student's two-tailed *t* test). **(C and D)** Representative images of RPE-1 cells stably expressing GFP-only, GFP-LUZP1, GFP-EPLINα, or GFP-EPLINβ and stained with antibodies against PCNT (centrosome marker) and ARP2 (C), or with phalloidin (D). **(E)** Western blot analysis of endogenous and LUZP1 and EPLIN in RPE-1 cells transfected with the indicated siRNAs for 72 h. **(F)** WT RPE-1 cells were transfected for 72 h with a control siRNA or siRNAs targeting LUZP1 or EPLIN, individually or in combination. Bar graph shows the mean percentage of ciliated cells (*n* > 200 cells per sample, three independent experiments) in each population. Error bars indicate SD. *, P < 0.05 (Student's two-tailed *t* test). **(G)** IF analysis of RPE-1 GFP-EPLINβ cells transfected for 72 h with control siRNA or siRNAs targeting LUZP. During the last 16 h of transfection, cells were treated with either DMSO (control) or 50 nM cytochalasin D. Cells were stained for γ-tubulin (centrosome and basal body marker) and ARL13B (ciliary marker). DNA was stained with DAPI. **(H)** Western blot analysis using protein extracts prepared from RPE-1 control cell lines and lines expressing GFP-LUZP1 or GFP-EPLINα or β. The cells were transfected with a control siRNA or siRNAs targeting LUZP1 or EPLIN, as indicated. A GFP antibody was used to detect the GFP-fusions and antibodies against LUZP1 and EPLIN were used to detect the respective endogenous proteins.

Video 1.   **LUZP1 localizes to actin filaments, centrosomes, and the midbody.** RPE-1 cells stably expressing GFP-LUZP were imaged every 5 min under controlled temperature and $CO_2$ growth conditions. Arrowheads point to centrosomes in interphase and mitotic cells, and the arrow points to the midbody. The video is played at 7 frames per second.

**Tables S1 and S2 are provided online. Table S1 contains the BioID-MS data. Table S2 lists the primers, siRNAs, and antibodies used in this work.**

