## [Peer Review File · The Journal of Cell Biology]

LUZP1 and the tumour suppressor EPLIN modulate actin stability to restrict primary cilia formation

João Gonçalves, Amit Sharma, Etienne Coyaud, Estelle Laurent, Brian Raught, and Laurence Pelletier

Corresponding Author(s): Laurence Pelletier, Lunenfeld-Tanenbaum Research Institute and João Gonçalves, Lunenfeld-Tanenbaum Research Institute

Review Timeline:	Submission Date:	2019-08-15
	Editorial Decision:	2019-09-21
	Revision Received:	2020-03-11
	Editorial Decision:	2020-03-27
	Revision Received:	2020-03-31

Monitoring Editor: Maxence Nachury

Scientific Editor: Marie Anne O'Donnell

Transaction Report:

DOI: <https://doi.org/10.1083/jcb.201908132>

September 20, 2019

Re: JCB manuscript #201908132

Dr. Laurence Pelletier
Lunenfeld-Tanenbaum Research Institute
600 University Avenue
Toronto M5G 1X5
Canada

Dear Dr. Pelletier,

Thank you for submitting your manuscript entitled "LUZP1 and the tumour suppressor EPLIN are negative regulators of primary cilia formation". The manuscript has been evaluated by expert reviewers, whose reports are appended below. Unfortunately, after an assessment of the reviewer feedback, our editorial decision is against publication in JCB.

You will see that there is interest in the identification of actin regulators that affect ciliogenesis, but the reviewers conclude the data are not sufficient to support the model proposed for the activity of LUZP1 and EPLIN. In particular, it would be necessary to demonstrate more directly that LUZP1 regulates ciliogenesis at the centrosome rather than indirectly from the myriad of other locations for LUZP1 in the cell. No specific experimental strategy is suggested by the reviewers, and we appreciate this is clearly not a straightforward question to address, but without some evidence for a centrosomal function of LUZP1, the advance is not considered appropriate for publication at the JCB. In addition, much of the imaging needs to be repeated with endogenous proteins or with these proteins expressed at near-endogenous levels.

If you have a clear idea for how to demonstrate a role of LUZP1 at the centrosome, we would be willing to consider a revision. But, we are cognizant of the difficulties of making a persuasive case for such a broadly distributed protein having specific functions at the centrosome, and therefore would encourage you to consider alternative paths to publication.

Although your manuscript is intriguing, I feel that the points raised by the reviewers are more substantial than can be addressed in a typical revision period. If you wish to expedite publication of the current data, it may be best to pursue publication at another journal.

Given interest in the topic, I would be open to resubmission to JCB of a significantly revised and extended manuscript that fully addresses the reviewers' concerns and is subject to further peer-review. If you would like to resubmit this work to JCB, please contact the journal office to discuss an appeal of this decision or you may submit an appeal directly through our manuscript submission system. Please note that priority and novelty would be reassessed at resubmission.

Regardless of how you choose to proceed, we hope that the comments below will prove constructive as your work progresses. We would be happy to discuss the reviewer comments further once you've had a chance to consider the points raised in this letter. You can contact the journal office with any questions, cellbio@rockefeller.edu or call (212) 327-8588.

Thank you for thinking of JCB as an appropriate place to publish your work.

Sincerely,

Maxence Nachury, Ph.D.
Monitoring Editor

Marie Anne O'Donnell, Ph.D.
Scientific Editor

Journal of Cell Biology

Reviewer #1 (Comments to the Authors (Required)):

This manuscript uses BioID, microscopy, pull-down assays, and RNAi knockdown methods to examine a possible role for LUZP1 and EPLIN in formation of primary cilia and regulation of their length. For the reasons outlined below, the results fail to support the primary conclusion stated in the title, that the two proteins are negative regulators of primary cilia formation. In addition, the manuscript presents results on multiple proteins, cellular structures, and possible protein interactions, but doesn't arrive at a coherent, compelling message.

1. Although the evidence does show that cells exhibited altered ciliogenesis after RNAi experiments that caused reduced levels of overexpressed LUZP1 and EPLIN, evidence is lacking that the alterations of ciliogenesis were due to a direct role of the proteins in ciliogenesis. As the manuscript indicates, both proteins interact with the cytoskeleton and are present throughout cells. Alteration of their levels almost certainly will alter many cellular properties, making it impossible to determine whether the effects on ciliation are direct or indirect.
2. Supplemental Figure 3B is described as presenting information on cell proliferation, yet, the Y-axis is labeled Phase Object Confluence (percent), with no explanation of the meaning of this term and its relationship to cell proliferation.
3. The manuscript lacked any information about the expression or location of the endogenous proteins, or of tagged proteins expressed at endogenous levels. Depending solely on overexpressed proteins to evaluate protein properties and their cellular roles can lead to erroneous conclusions.
4. Potential interactions between overexpressed LUZP1 and EPLIN are described in the manuscript, but the importance in ciliogenesis, the main topic of this manuscript, of those possible interactions are not addressed. In addition, overexpressed LUZP1 is present at centrosomes, but overexpressed EPLIN is not. The manuscript lacks any discussion of likely interpretations of these results.

Reviewer #2 (Comments to the Authors (Required)):

Gonçalves et al. show that actin-binding protein, LUZP1, acts as a negative regulator of primary cilia formation. They first demonstrated that GFP-LUZP1 is localized to F-actin and centrosome through its C-terminal domain. To investigate its function, they performed proximity-labeling proteomics and

showed the list of potential interaction partners of LUZP1. Combining this and other previous proteomics reports, they focused on EPLIN. GFP-EPLIN is localized to actin-based structures, but not to centrosome. They found that LUZP1 downregulation by siRNA induces aberrant ciliogenesis in proliferating cells. A similar result was observed in EPLIN knockdown experiments. Finally, they found that LUZP1 and EPLIN overexpression can enhance actin polymerization. Although the results are interesting, there are several issues that need to be addressed.

Major comments:

1. How LUZP1 and EPLIN regulate actin dynamics for ciliogenesis remains unknown. Without this key information, justification to focus on these two molecules in the study is weak. For example, can downregulation of LUZP1 and/or EPLIN change the pattern of actin polymerization around the basal body or primary cilia? Also, some actin regulators (Arp3/N-WASP) are localized to basal body (Drummond, et al. J. Cell Biol. 217, 3255). Are LUZP1 and EPLIN involved in the subcellular localization of these actin regulators?
2. Along with the point raised in #1, a lack of the following information reduces the significance of the work: a functional relationship between LUZP1 and EPLIN (besides their physical binding), and localization of "endogenous" proteins in wild type and knockdown cells.
3. While the authors demonstrated that LUZP1/EPLIN downregulation induces ciliogenesis, whether overexpression of LUZP1/EPLIN reduces ciliogenesis in serum-starved cells is not examined.
4. Difference/comparison between this work and Uddin et al. 2019 EMBO should be clearly described.
5. The authors claim that the proximity labeling detects FLNA as an interaction partner of LUZP1, so that the authors decided to characterize EPLIN, a known interaction partner of FLNA. However, FLNA could be a false-positive. Validating that BirA-only control cannot biotinylate FLNA would be necessary. Alternatively, interaction between FLNA and LUZP1 should be pursued.
6. The N-terminus of LUZP1 has not been characterized well. Provided its irrelevance to the protein localization (Fig. 1C) and to the interaction with EPLIN (Fig. S2D), how is this region involved in the protein function? For example, can the N-terminal truncate rescue the phenotype shown in Fig. 3B?

Minor comments:

1. The fluorescence intensity of dye-labeled phalloidin is used as an indicator of the extent of actin polymerization, which should be quantified.
2. Typos
 - "a MT" should be "an MT"
 - "Here we show that LUZP1 is as an actin..." should be "Here we show that LUZP1 is an actin..."
 - "phalloidin co-staining (Fig. 1C, D)" should be "phalloidin co-staining (Fig. 1D)"
 - "cytochalasin D lead to" should be "cytochalasin D led to"

Reviewer #3 (Comments to the Authors (Required)):

In this manuscript, Joao Goncalves and colleagues studied the role of LUZP1, which they had previously identified as a centriolar protein involved in the regulation of ciliogenesis (Gupta Cell 2015). They found that it binds actin filaments and thereby modulates ciliogenesis. They also found that LUZP1 interacts with EPLIN and that EPLIN also regulates the actin-dependent ciliogenesis pathway. Surprisingly it seemed that the way LUZP1 and EPLIN modulate ciliogenesis can be at least partially independent.

This work shine some light on the role of actin in ciliogenesis. It is timely and important since we don't know much about this regulation although there are accumulating evidences that actin is a major regulator of ciliogenesis. The work is solid and well constructed. Many important controls have

been performed in order to properly describe the specific contributions of LUZP1 and EPLIN to the actin-dependent ciliogenesis pathways. But it remains unclear whether these proteins regulate ciliogenesis by modulating actin at the centrosome. They are clearly centrosome/cilia associated proteins. And they appeared capable to interfere with the way actin disassembly promote ciliogenesis. But these proteins could modulate actin network at distance from the centrosome yet impact ciliogenesis and cilium length. The authors did pay attention not to make any overstatements in the text about this remaining uncertainty. But it is a missing piece in the picture and the presentation of the work can be considered confusing about it.

I can appreciate how difficult it is to distinguish the centrosome-associated effects from the others. But any progress in that direction would better connect the different parts of the study and definitely improved the overall strength of the paper.

Considering authors' expertise in super-resolution, it is surprising not to see well contrasted images of actin at the centrosome and the localisation of LUZP1/EPLIN in this centrosomal network. RPE1 may not be the most appropriated system to visualize centrosomal actin though. Indeed centrosomal actin has recently been shown to be negatively correlated to the extent of cell spreading. It may be worth looking at the relationship between centrosomal actin, LUZP1 concentration at the centrosome and ciliogenesis in dense monolayer of epithelial cells.

We would like to thank the three referees for their enthusiasm, judicious comments and thoughtful suggestions about our work. Below is our detailed point-by-point response. Our responses are **in bold** and the original comments in their entirety are *in italics*. The referee's reports have been very helpful and we hope that the reviewers will find our revised manuscript suitable for publication as a short report in the *Journal of Cell Biology*.

Reviewer #1

This manuscript uses BioID, microscopy, pull-down assays, and RNAi knockdown methods to examine a possible role for LUZP1 and EPLIN in formation of primary cilia and regulation of their length. For the reasons outlined below, the results fail to support the primary conclusion stated in the title, that the two proteins are negative regulators of primary cilia formation. In addition, the manuscript presents results on multiple proteins, cellular structures, and possible protein interactions, but doesn't arrive at a coherent, compelling message.

1. Although the evidence does show that cells exhibited altered ciliogenesis after RNAi experiments that caused reduced levels of overexpressed LUZP1 and EPLIN, evidence is lacking that the alterations of ciliogenesis were due to a direct role of the proteins in ciliogenesis. As the manuscript indicates, both proteins interact with the cytoskeleton and are present throughout cells. Alteration of their levels almost certainly will alter many cellular properties, making it impossible to determine whether the effects on ciliation are direct or indirect.

We thank this referee for her/his comments and apologise if the title we originally chose was in any way misleading, which was certainly not our intention. As mentioned by this reviewer, we indeed concluded from the functional assays we performed (loss of function and over-expression) that LUZP1 and EPLIN are negative regulators of primary cilia formation. Our work is by no mean the first to show effects of actin network perturbations on the ability of cells to ciliate. For example, Kim and colleagues performed an RNAi screen and identified a number of actin-related proteins that act as negative regulators of primary cilia formation and length (Kim et al., 2010). In the original version of our manuscript we presented data, data which is also supported by very recent work from others, suggesting that the increase in ciliation caused by LUZP1 and EPLIN depletion in RPE-1 cells is related to their role in actin stabilization (Bozal-Basterra L., 2019). Having said this, the reviewer's point is well taken and for the reasons discussed below, and in light of our new data, we have changed our title to be less ambiguous and now use "LUZP1 and the tumour suppressor EPLIN modulate actin nucleation at the centrosome to restrict primary cilia formation".

In order to further investigate the mechanisms underpinning the role of these proteins in cilia formation specifically at the centrosome, we conducted a number of loss of function and over-expression experiments in RPE-1 and showed that LUZP1 and EPLIN depletion lead to an increase in the levels of MyosinVa at the centrosome (Fig. 4A, B), the site where primary cilia formation is initiated. Interestingly, the accumulation of MyosinVa at the centrosome has been shown to be required for the formation of the ciliary vesicle, one of the earliest steps of cilia formation (Wu et al., 2018). Moreover, ciliation induced by cytochalasin D treatment was recently shown to be due to the accumulation of MyosinVa at

the centrosome (Wu et al., 2018). We now further show that over-expression of LUZP1 and EPLIN leads to increased levels of the actin nucleator ARP2 and actin at the centrosome (Sup. Fig. 3 A-D), which causes a small decrease in ciliation upon their overexpression in RPE-1 cells (Sup. Fig. 2F). Together, these new results, combined with our previous observations and those of others, supports a model whereby LUZP1 and EPLIN actin-related functions are important in cilia formation through their role in the regulation of the levels of actin-associated proteins, and actin at the centrosome, which in turn impacts the formation of primary cilia. We agree with this reviewer that our results do not directly implicate LUZP1 in EPLIN in cilia formation, for example like distal appendages and IFT defects do, but they nonetheless provide some mechanistic insight on how actin regulation at the centrosome participates in this process. This is now discussed better on pages 13 and 14 of the revised manuscript.

2. Supplemental Figure 3B is described as presenting information on cell proliferation, yet, the Y-axis is labeled Phase Object Confluence (percent), with no explanation of the meaning of this term and its relationship to cell proliferation.

We apologize for this. We have changed the figure legend and the respective section in the material and methods to clarify that what we measured was cell confluence as a proxy for proliferation using an IncuCyte® Live Cell Analysis Instrument.

3. The manuscript lacked any information about the expression or location of the endogenous proteins, or of tagged proteins expressed at endogenous levels. Depending solely on overexpressed proteins to evaluate protein properties and their cellular roles can lead to erroneous conclusions.

We thank the reviewer pointing out this shortcoming. Initially, the characterization of LUZP1 endogenous localization was not possible due to the lack of a commercially available antibody that worked by IF. Therefore, we had to study the localization of the protein by expressing it fused to GFP, mCherry, FLAG and FLAG-BirA*. Furthermore, there were no EPLIN antibodies available that could discriminate between its two isoforms. This led us to analyse the localization of EPLIN α and β in stable lines expressing GFP-tagged versions. To address this reviewers' concern, we have now performed IF analysis using a recently developed LUZP1 antibody, and EPLIN antibodies (Fig. 1 B-D; Fig. 2 F; Sup. Fig. 2 F). This analysis confirmed that endogenous LUZP1 localizes to the centrosome, the basal body of primary cilia, actin filaments and the midbody, in line with results from others (Bozal-Basterra L., 2019). Moreover, the observed EPLIN signal was consistent with the one observed for GFP-EPLIN α and the fact that this is the main isoform expressed in RPE-1 cells (Fig. 2 C, F). Although we have confirmed the localization of both LUZP1 and EPLIN using antibodies that detect the endogenous proteins, the fluorescently-tagged variants of LUZP1 and EPLIN remain an important tool for our study in a number of ways: i) they were used to characterize our stable cells lines; ii) they were used to define which LUZP1 region is required for its basal-body localization; iii) they were used to determine where EPLIN α and β localize in RPE-1 cells which is not possible using antibodies due to lack of isoform specific ones. We thus believe that the analysis of the endogenous proteins combined with that of their tagged counterparts now provides a compelling picture of their sub-cellular localizations.

This reviewer also wanted to know if EPLIN and LUZP1 were expressed in these cells. To address this, we provide western blot analyses of endogenous LUZP1 and EPLIN in RPE-1 and HEK293T cells showing that the proteins are indeed expressed in these cell lines (Fig. 3 C, F; Sup. Fig. 2 A-C; Sup. Fig. 3 F, H). In addition, the expression of EPLIN isoforms was also investigated in other cell lines (U2-OS and MCF-7; Fig. 2 C).

4. Potential interactions between overexpressed LUZP1 and EPLIN are described in the manuscript, but the importance in ciliogenesis, the main topic of this manuscript, of those possible interactions are not addressed. In addition, overexpressed LUZP1 is present at centrosomes, but overexpressed EPLIN is not. The manuscript lacks any discussion of likely interpretations of these results.

We thank the reviewer for this comment, and we agree with that we did not fully clarify the relevance of the interaction between LUZP1 and EPLIN and its effect on ciliogenesis. We have validated the interaction between LUZP1 and EPLIN to the best of our abilities. In the manuscript, we report results of semi-endogenous co-immunoprecipitations in two different cell lines (RPE-1 and HEK293) showing that tagged LUZP1 can pull down endogenous EPLIN and tagged EPLIN pulls down endogenous LUZP1 (Fig. 2D, E; Sup. Fig. 2A, B). The only case in which both proteins were tagged corresponds to the experiment in which we tested the interaction between EPLIN and LUZP1 truncation mutants (Sup. Fig. 2 B). The use of epitope tags for the pull-downs was necessary as the cells do not express the mutants endogenously. We also discuss in the manuscript that the interaction between LUZP1 and EPLIN had already been observed in another study performed in HeLa cells and that both proteins are part of the BioID interactome of CDC14A, a phosphatase that regulates actin organization through dephosphorylation of EPLIN, and also regulates primary cilia length (Uddin et al., 2019). This interaction has therefore been detected by distinct mass-spectrometry approaches and validated extensively by semi-endogenous Co-IPs.

We have now confirmed that LUZP1 and EPLIN localize to actin filaments but tend to accumulate in distinct sub-cellular domains and structures (Fig. 2 F). EPLIN localizes to the leading edge of the cell whereas LUZP1 accumulates at the rear end (Fig. 2 F). Also, LUZP1 localizes to the centrosome/basal body and the midbody whereas EPLIN does not (Fig. 1 B-D; Fig. 2 G; Sup. Fig 1 A, B, C, F, G). These different localizations are strong indications that the two proteins may share some functions but likely have additional roles. Dissecting the shared and independent functions of LUZP1 and EPLIN is critical but entails a significant amount of work which is beyond the scope of the manuscript. Nevertheless, we provide several data from loss of function and over-expression experiments suggesting a functional relationship between these two proteins, which seem to participate in the modulation of actin and actin regulators at the centrosome, impacting the early steps of primary cilia formation.

Reviewer #2

Gonçalves et al. show that actin-binding protein, LUZP1, acts as a negative regulator of primary cilia formation. They first demonstrated that GFP-LUZP1 is localized to F-actin and centrosome

through its C-terminal domain. To investigate its function, they performed proximity-labeling proteomics and showed the list of potential interaction partners of LUZP1. Combining this and other previous proteomics reports, they focused on EPLIN. GFP-EPLIN is localized to actin-based structures, but not to centrosome. They found that LUZP1 downregulation by siRNA induces aberrant ciliogenesis in proliferating cells. A similar result was observed in EPLIN knockdown experiments. Finally, they found that LUZP1 and EPLIN overexpression can enhance actin polymerization. Although the results are interesting, there are several issues that need to be addressed.

Major comments:

1. How LUZP1 and EPLIN regulate actin dynamics for ciliogenesis remains unknown. Without this key information, justification to focus on these two molecules in the study is weak. For example, can downregulation of LUZP1 and/or EPLIN change the pattern of actin polymerization around the basal body or primary cilia? Also, some actin regulators (Arp3/N-WASP) are localized to basal body (Drummond, et al. *J. Cell Biol.* 217, 3255). Are LUZP1 and EPLIN involved in the subcellular localization of these actin regulators?

We thank the reviewer for this comment, which was also raised by reviewer #1 as discussed above. To address this important point, we conducted several experiments aimed at investigating the impact of LUZP1 and EPLIN loss of function or over-expression on actin regulators at the centrosome, and how this may influence ciliation. By silencing LUZP1 and EPLIN we observed increased levels of MyosinVa at the centrosome (Fig. 5 A, B). MyosinVa is a key player in the early steps of cilia formation, participating in the formation of the ciliary vesicle (Wu et al., 2018). These results, suggest that, similar to cytochalasin D treatment, depleting these proteins disrupts actin and that leads to the accumulation of MyosinVa at the centrosome facilitating ciliogenesis (Wu et al., 2018). This hypothesis is supported by our results showing that over-expressing LUZP1 or EPLIN can rescue the effect of cytochalasin D on ciliation (Fig. 4D). In addition, we observed increased levels of the actin nucleator ARP2 and actin itself at the centrosome in our stable lines over-expressing LUZP1 or EPLIN. Concomitantly, our LUZP1 and EPLIN over-expressing lines show a slightly reduced ability to ciliate (Sup. Fig. 2 F). We believe these new data provide some insight into mechanisms by which LUZP1 and EPLIN impact primary cilia formation. Overall, these and other results presented in the manuscript show that LUZP1 is a novel actin regulator and that its role in ciliation, as well as that of EPLIN, is related to actin-associated functions. Furthermore, a recent report on bioRxiv showed that LUZP1 loss of function in NIH3T3 and human patient derived cells causes an overall decrease in actin filaments in the cell (Bozal-Basterra et al. 2019) and an increase in ciliation, supporting a role for LUZP1 as a stabilizer of actin and a negative regulator of cilia formation.

2. Along with the point raised in #1, a lack of the following information reduces the significance

of the work: a functional relationship between LUZP1 and EPLIN (besides their physical binding), and localization of "endogenous" proteins in wild type and knockdown cells.

We thank the reviewer for raising this valid concern. Indeed, analysing endogenous proteins is always ideal but we were originally limited by the available tools, specifically antibodies. Fortunately, a recently developed LUZP1 antibody allowed us to investigate the localizations of the endogenous protein. With this analysis we confirmed that endogenous LUZP1 localizes to actin filaments, the centrosome, the basal body and the midbody (Fig. 1B-D). Analysis of endogenous EPLIN showed the protein at actin filaments but also accumulating at filopodia in the leading edge of the cell (Fig. 2 F; Sup. Fig. 2 F), similar to what was observed for EPLIN α , and is consistent with this being the predominant isoform in RPE-1 cells (Fig. 2 C). The analysis of the endogenous proteins, combined with our previous analysis using tagged proteins, allowed us to observe that LUZP1 and EPLIN can co-localize to actin filaments but clearly accumulate in different cellular domains and actin structures (Fig. 2F). Moreover, only LUZP1 localizes to the centrosome and the basal body. In fact, this distinct localization patterns are a good indication that, although LUZP1 and EPLIN may share some functions, these proteins likely also have distinct roles.

We chose to follow up on the interaction between LUZP1 and EPLIN for several reasons: i) the interaction was identified in our LUZP1 BioID interactome, and in a large-scale mass-spectrometry study in HeLa cells (Hein et al., 2015); ii) both proteins were identified in the proximity interactome (BioID) of the phosphatase CDC14A (a regulator of actin and cilia length) (Uddin et al., 2019); iii) EPLIN is a well-studied actin stabilizer (Maul et al., 2003). We went on to validate this interaction by semi-endogenous Co-IPs in two cell lines (Fig. 2D, E; Sup. Fig. 2 A, B). Moreover, we showed that the loss-of-function phenotypes for LUZP1 and EPLIN were very similar (ciliation increase, ciliary lengthening, and provide new data on MyosinVa accumulation at the centrosome), and that LUZP1 is needed for over-expressed EPLIN to rescue the ciliation induction caused by cytochalasin D (Fig. 5 F). Together, these data support the existence of a functional relationship between LUZP1 and EPLIN in the regulation of actin dynamics at the centrosome. Fully elucidating the molecular pathways these proteins are involved in will be an important future step. It appears likely that these proteins play multiple roles in actin-related or others process, potentially in different cellular locales. Nevertheless, dissecting LUZP1 and EPLIN functions (whether they are shared or independent of each other) is an exciting future endeavour which will entail a considerable amount of work, beyond the scope of this JCB report. This is now more clearly discussed on page 14 of the manuscript.

3. While the authors demonstrated that LUZP1/EPLIN downregulation induces ciliogenesis, whether overexpression of LUZP1/EPLIN reduces ciliogenesis in serum-starved cells is not examined.

This is a very valid point and we thank the reviewer for pointing it out. To address this concern, we analysed the ciliation ability of our RPE-1 stable lines and observed that there is a small but significant decrease in ciliation in the cells expressing GFP-LUZP1 and GFP-EPLIN (Sup. Fig. 2E). Of note, the cells were generated using a lentiviral system, which leads to stable but moderate levels of the fusion protein compared to what one would obtain by plasmid transfection and transient expression. This moderate effect on ciliation may be due to the observed increased levels of ARP2 and actin at the centrosome in these stable lines compared to control cells expressing GFP only (Sup. Fig. 3 A-D). These results were included in the manuscript and are now discussed on page 14.

4. Difference/comparison between this work and Uddin et al. 2019 EMBO should be clearly described.

This is an excellent suggestion. We now provide further discussion of the work by Uddin et al. on page 6 of the revised manuscript. Indeed, both EPLIN/LIMA1 and LUZP1 were shown to be proximal interactors of CDC14A and exploring the functional link between these proteins, in particular how LUZP1 and EPLIN activity is regulated during the cell cycle by phosphorylation and dephosphorylation, and how this impinges on ciliation is a very exiting area of future research.

5. The authors claim that the proximity labeling detects FLNA as an interaction partner of LUZP1, so that the authors decided to characterize EPLIN, a known interaction partner of FLNA. However, FLNA could be a false-positive. Validating that BirA-only control cannot biotinylate FLNA would be necessary. Alternatively, interaction between FLNA and LUZP1 should be pursued.

We apologize for not sufficiently detailing the bayesian approach used to filter out false positive proximity interactions. Our mass-spectrometry data was analysed by SAINT (Significance Analysis of INTeractome), a probabilistic method for scoring bait-prey interactions against negative controls. The SAINT algorithm uses spectral counts or protein intensities as the input to calculate the probability of true interaction, allowing for the selection of high-confidence interactions (Teo et al., 2016). Additional information on SAINT was included in the methods section of the paper.

Indeed, as pointed out by the reviewer, we did identify FLNA as a high confidence proximity interactor of LUZP1. In our BioID studies we used HEK293 cells expressing FLAG-BirA only as controls and all the data is presented in Fig. 2 A and Sup. Table 1. FLNA peptides were detected in the control samples but there were significantly more in the FLAG-BirA-LUZP1 samples therefore having been considered a strong interactor by SAINT analysis. The interaction between LUZP1 and FLNA was also reported and studied

extensively in a recently published article (Wang and Nakamura, 2019) which we discuss on page 6 of our manuscript.

Although our proximity interaction mapping of LUZP1 had provided us with a number of putative interactions to follow up on, we opted to follow up on the EPLIN-LUZP1 interaction for multiple reasons: i) EPLIN scored as a strong interactor in our LUZP1 BioID data (Fig. 2 A); ii) The LUZP1-EPLIN interaction was detected in a large mass-spectrometry study in HeLa cells (Hein et al., 2015); iii) both proteins were identified in the proximity interactome (BioID) of the phosphatase CDC14A (regulator of actin and cilia length) (Uddin et al., 2019); iv) and EPLIN is a well-studied actin stabilizer (Maul et al., 2003). Of note FLNA has been implicated in cilia formation (required for basal body docking) and although it surely remains of interest exploring further deciphering the potential functional relationship between the LUZP1-FLNA interaction in the context of ciliation likely fall beyond the scope of this report.

6. *The N-terminus of LUZP1 has not been characterized well. Provided its irrelevance to the protein localization (Fig. 1C) and to the interaction with EPLIN (Fig. S2D), how is this region involved in the protein function? For example, can the N-terminal truncate rescue the phenotype shown in Fig. 3B?*

The structure/function analysis of LUZP1 (leucine zipper protein 1) is indeed a critical issue to understand this still poorly understood protein in terms of the domains responsible for its localizations and functions. In our manuscript we have shown that the only ascribed domains present in LUZP1 (at the N-terminus), which give the protein its name, are not required for the localization to actin and the centrosome or the interaction with EPLIN.

A recent report by Wang and Nakamura extensively addressed these issues by analysing the localization of multiple LUZP1 fragments as well as their ability to bind to and promote cross-linking of actin *in vitro*. This study, however, did not report LUZP1 localizations to the centrosome, basal body or midbody. These authors showed, for example, that a LUZP1 fragment comprised of amino acids (aa) residues 1-500 localizes to actin stress fibers in MEFs whereas a fragment comprised of aa 1-400 does not, showing that there is at least one actin-binding domain in the region between aa 400-500. This aa 400-500 domain was indeed proven to be required for actin binding but not sufficient to cause actin cross-linking (Wang and Nakamura, 2019). In agreement, we showed that the over-expression of a truncation mutant of LUZP1 (aa 1-496) in RPE-1 cells causes a dramatic effect on actin filaments which appear curved, matching the observation that the aa 1-500 fragment causes actin bundling *in vitro*. Furthermore, it was shown that a fragment comprised of aa 1-359 does not localize to actin (matching our results), it oligomerizes, and is necessary to elicit actin cross-linking (Wang and Nakamura, 2019).

Despite this extensive biochemical characterization of the LUZP1 protein, many questions still need to be answered regarding what determines its several distinct localizations, if the full length protein works as an oligomer, what domains determine the interactions with its partners and are responsible for its function(s). Using multiple mutants in order to try to rescue specific phenotypes such as ciliation is an approach that needs to be taken with considerable care as we still have to understand how they behave compared to the full-length protein, for example if they behave as monomers or oligomers, how they may interact with distinct LUZP1 partners, where they localize etc. Also, as mentioned above, the expression of LUZP1 truncation mutants can cause dramatic phenotypes in terms of actin organization which likely impact ciliation and will require engineering cell lines, most likely through genome editing to avoid overexpression and related issues. The detailed LUZP1 structure-function analysis is part of a bigger project ongoing on in the lab and represents a considerable amount of work which we feel falls beyond the scope of this short report.

Minor comments:

1. *The fluorescence intensity of dye-labeled phalloidin is used as an indicator of the extent of actin polymerization, which should be quantified.*

We thank the reviewer for this comment. Accordingly, such quantifications showing higher phalloidin fluorescence levels in cells over-expressing LUZP1 or EPLIN compared to neighboring non-expressing cells have been added to the respective figures (Fig. 4A, B).

2. *Typos:*

- *"a MT" should be "an MT"*

"Here we show that LUZP1 is as an actin..." should be "Here we show that LUZP1 is an actin..."

- *"phalloidin co-staining (Fig. 1C, D)" should be "phalloidin co-staining (Fig. 1D)"*
- *"cytochalasin D lead to" should be "cytochalasin D led to"*

These typos were corrected.

Reviewer #3 (Comments to the Authors (Required)):

In this manuscript, Joao Goncalves and colleagues studied the role of LUZP1, which they

had previously identified as a centriolar protein involved in the regulation of ciliogenesis (Gupta Cell 2015). They found that it binds actin filaments and thereby modulates ciliogenesis. They also found that LUZP1 interacts with EPLIN and that EPLIN also regulates the actin-dependent ciliogenesis pathway. Surprisingly it seemed that the way LUZP1 and EPLIN modulate ciliogenesis can be at least partially independent

This work shines some light on the role of actin in ciliogenesis. It is timely and important since we don't know much about this regulation although there are accumulating evidences that actin is a major regulator of ciliogenesis. The work is solid and well constructed. Many important controls have been performed in order to properly describe the specific contributions of LUZP1 and EPLIN to the actin-dependent ciliogenesis pathways. But it remains unclear whether these proteins regulate ciliogenesis by modulating actin at the centrosome. They are clearly centrosome/cilia associated proteins. And they appeared capable to interfere with the way actin disassembly promote ciliogenesis. But these proteins could modulate actin network at distance from the centrosome yet impact ciliogenesis and cilium length. The authors did pay attention not to make any overstatements in the text about this remaining uncertainty. But it is a missing piece in the picture and the presentation of the work can be considered confusing about it.

I can appreciate how difficult it is to distinguish the centrosome-associated effects from the others. But any progress in that direction would better connect the different parts of the study and definitely improved the overall strength of the paper.

Considering authors' expertise in super-resolution, it is surprising not to see well contrasted images of actin at the centrosome and the localisation of LUZP1/EPLIN in this centrosomal network. RPE1 may not be the most appropriated system to visualize centrosomal actin though. Indeed centrosomal actin has recently been shown to be negatively correlated to the extent of cell spreading. It may be worth looking at the relationship between centrosomal actin, LUZP1 concentration at the centrosome and ciliogenesis in dense monolayer of epithelial cells.

We thank the reviewer for their appreciation of our work, and the extremely valid points he/she raised. In order to address the reviewer's concerns and improve our manuscript we conducted the following experiments:

Recently developed antibodies were used to analyse the localizations of endogenous LUZP1 and EPLIN (Fig. 1 B-D; Fig. 2 F; Sup. Fig. 2 F). We confirmed that LUZP1 localizes to actin filaments, the centrosome, the basal body of primary cilia and the midbody. We also showed that the localization of endogenous EPLIN matches the one of GFP-EPLIN α , which agrees with this being the major isoform in RPE-1 cells (Fig. 2 C, F). Most importantly, our combined sub-cellular localization analysis shows that LUZP1 and EPLIN can co-localize at actin filaments, but clearly accumulate at different cellular domains (Fig. 2F). EPLIN accumulates at membrane ruffles at the leading edge whereas

LUZP1 accumulates at the opposite end of the cell. Moreover, EPLIN does not localize to the centrosome, the basal body or the midbody (Fig. 2F, G; Sup Fig. 2 F, G). This strongly suggests that the two proteins may share some functions but likely also have distinct roles, actin-related or others. Nevertheless, the loss of function of both LUZP1 and EPLIN caused similar effects on ciliation and cilia length. Moreover, their over-expression rescues the induced ciliation caused by cytochalasin D, suggesting that both LUZP1 and EPLIN actin-stabilization roles impact cilia formation.

To further elucidate the mechanisms by which LUZP1 and EPLIN depletion induces ciliation we studied the levels of MyosinVa at the centrosome in RPE-1 LUZP1 or EPLIN-depleted cells and showed this protein accumulates at the centrosome following RNAi treatment (Fig. 5A, B). MyosinVa accumulation was also shown to happen upon cytochalasin D treatment, leading to increased ciliation (Wu et al., 2018). Indeed, MyosinVa is involved in the early steps of ciliogenesis by participating in the formation of the ciliary vesicle. This result is also in agreement with the fact that over-expressing LUZP1 and EPLIN rescues the induced ciliation caused by cytochalasin D. Furthermore, we showed that the levels of the actin nucleator ARP2 as well as that of actin are higher in our stable lines over-expressing GFP-tagged LUZP1 or EPLIN, compared to those in control GFP-only cells (Sup. Fig. 3 A, B). This increase in centrosomal ARP2 and actin might be responsible for the small decrease in ciliation observed in LUZP1 and EPLIN over-expressing lines (Sup. Fig. 2F). Overall, these results show that manipulating the levels of LUZP1 and EPLIN affects the levels of actin-associated proteins at centrosome, with possibly impacts actin nucleation and dynamics at this organelle and ciliogenesis.

We also make note in the revised version of the manuscript that in a preprint on BioRxiv it was reported that the depletion of LUZP1 causes an overall decrease in actin filaments in NIH3T3 LUZP1 null cells, as well as LUZP1 mutant patient derived cells (Bozal-Basterra et al. 2019). Moreover, this study also shows that LUZP1 is a negative regulator of primary cilia formation. These results further validate and complement the observations reported in our manuscript.

Finally, we appreciate the suggestion that super-resolution imaging would provide more detailed information regarding the localization of LUZP1 and its relationship with actin at the centrosome. We have tried very hard over the last 3 months to get this to work, and although we have been very successful in the past doing these types of analyses on PCM, centriole and centriolar satellite organization, we have failed spectacularly when we attempted this with actin and LUZP1 at the centrosome using either antibodies to the endogenous proteins or cell lines expressing GFP variants of the given proteins. We remain unsure why we were unable to get this to work but we can speculate on a number of possibilities. As mentioned by the referee, actin is not so easy to observe at centrosome. We attempted to do this in different cell types including keratinocytes and lymphocytes with no real success. Despite our best efforts we believe that with the reagents currently available

for these proteins, we are not able to reach high enough signal to noise ratios to generate artifact-free 3D-SIM reconstructions. We are disappointed not to have been able to satisfy this reviewer on this point but we nonetheless mention in the manuscript that it will be important in the future to explore actin organization and LUZP1 localisation at the sub-diffraction level.

References:

- Bozal-Basterra L., G.-S.M., Bermejo-Arteagabeitia A., Da Fonseca C., Pampliega O., Andrade R., Martín-Martín N, Branon T.C., Ting A.Y., Carracedo A., Rodríguez J.A., Elortza F., Sutherland J. D., Barrio R. 2019. LUZP1, a novel regulator of primary cilia and the actin cytoskeleton, is altered in Townes-Brocks Syndrome. *bioRxiv*.
- Hein, M.Y., N.C. Hubner, I. Poser, J. Cox, N. Nagaraj, Y. Toyoda, I.A. Gak, I. Weisswange, J. Mansfeld, F. Buchholz, A.A. Hyman, and M. Mann. 2015. A human interactome in three quantitative dimensions organized by stoichiometries and abundances. *Cell*. 163:712-723.
- Kim, J., J.E. Lee, S. Heynen-Genel, E. Suyama, K. Ono, K. Lee, T. Ideker, P. Aza-Blanc, and J.G. Gleeson. 2010. Functional genomic screen for modulators of ciliogenesis and cilium length. *Nature*. 464:1048-1051.
- Maul, R.S., Y. Song, K.J. Amann, S.C. Gerbin, T.D. Pollard, and D.D. Chang. 2003. EPLIN regulates actin dynamics by cross-linking and stabilizing filaments. *The Journal of cell biology*. 160:399-407.
- Teo, G., H. Koh, D. Fermin, J.P. Lambert, J.D. Knight, A.C. Gingras, and H. Choi. 2016. SAINTq: Scoring protein-protein interactions in affinity purification - mass spectrometry experiments with fragment or peptide intensity data. *Proteomics*. 16:2238-2245.
- Uddin, B., P. Partscht, N.P. Chen, A. Neuner, M. Weiss, R. Hardt, A. Jafarpour, B. Hessling, T. Ruppert, H. Lorenz, G. Pereira, and E. Schiebel. 2019. The human phosphatase CDC14A modulates primary cilium length by regulating centrosomal actin nucleation. *EMBO reports*. 20.
- Wang, J., and F. Nakamura. 2019. Identification of Filamin A Mechanobinding Partner II: Fimbacin Is a Novel Actin Cross-Linking and Filamin A Binding Protein. *Biochemistry*.
- Wu, C.T., H.Y. Chen, and T.K. Tang. 2018. Myosin-Va is required for preciliary vesicle transportation to the mother centriole during ciliogenesis. *Nature cell biology*. 20:175-185.

March 27, 2020

RE: JCB Manuscript #201908132R-A

Dr. Laurence Pelletier
Lunenfeld-Tanenbaum Research Institute
600 University Avenue
Toronto M5G 1X5
Canada

Dear Dr. Pelletier:

Thank you for submitting your revised manuscript entitled "LUZP1 and EPLIN modulate actin nucleation at the centrosome to restrict primary cilia formation". We would be happy to publish your paper in JCB provided the text and perhaps the title is amended to temper down the claims that the proteins under study are acting specifically at the centrosome, as recommended by Reviewer #3 and pending final revisions necessary to meet our formatting guidelines (see details below).

- Provide the main and supplementary texts as separate, editable .doc or .docx files
- Provide main and supplementary figures as separate, editable files according to the instructions for authors on JCB's website paying particular attention to the guidelines for preparing images and blots at sufficient resolution for screening and production
- Format references for JCB
- Add scale bars to figures S1C insets
- Add paragraph after the Materials and Methods section briefly summarizing all "Online Supplementary Materials"

A. MANUSCRIPT ORGANIZATION AND FORMATTING:

Full guidelines are available on our Instructions for Authors page, <http://jcb.rupress.org/submission-guidelines#revised>. **Submission of a paper that does not conform to JCB guidelines will delay the acceptance of your manuscript.**

B. FINAL FILES:

- An editable version of the final text (.DOC or .DOCX) is needed for copyediting (no PDFs).
- High-resolution figure and video files: See our detailed guidelines for preparing your production-

ready images, <http://jcb.rupress.org/fig-vid-guidelines>.

Thank you for this interesting contribution, we look forward to publishing your paper in Journal of Cell Biology.

Sincerely,

Maxence Nachury, Ph.D.
Monitoring Editor

Marie Anne O'Donnell, Ph.D.
Scientific Editor

Journal of Cell Biology

Reviewer #2 (Comments to the Authors (Required)):

All of the points from my initial review have been properly addressed.

Reviewer #3 (Comments to the Authors (Required)):

The authors did their best to address the concerns of the reviewers regarding the lack of clarity about the mechanism by which LUZP1 and EPLIN affect actin assembly at the centrosome and ciliogenesis.

They added some interesting and convincing data showing that the two proteins affect the centrosomal localisation of myosinIV, a key regulator of ciliary vesicle formation (new figure 5A). Furthermore authors added some data showing that the proteins affect the level of Arp2/3 and actin at the centrosome (new Sup Fig 3A-3D), although the stainings are not fully convincing

(reason why images are in supplementary data i guess).

Altogether, these data improves significantly the mechanistic insight in the mechanism of action of the to proteins.

It is still unclear to me if the proteins modulates actin directly act at the centrosome (unlikely for EPLIN ...) or somewhere else in the actin network with some indirect impact of the centrosomal network (as it has been shown by Inoue et al in EMBO Journal for the extent of cell adhesion for example). In that regard I find the title confusing since it somehow suggest that proteins act at the centrosome.